# Cell cycle entry triggers a switch between two modes of Cdc42 activation during yeast polarization

Kristen Witte, Devin Strickland, Michael Glotzer*

Department of Molecular Genetics and Cell Biology, University of Chicago, Chicago, United States

**Abstract** Cell polarization underlies many cellular and organismal functions. The GTPase Cdc42 orchestrates polarization in many contexts. In budding yeast, polarization is associated with a focus of Cdc42•GTP which is thought to self sustain by recruiting a complex containing Cla4, a Cdc42-binding effector, Bem1, a scaffold, and Cdc24, a Cdc42 GEF. Using optogenetics, we probe yeast polarization and find that local recruitment of Cdc24 or Bem1 is sufficient to induce polarization by triggering self-sustaining Cdc42 activity. However, the response to these perturbations depends on the recruited molecule, the cell cycle stage, and existing polarization sites. Before cell cycle entry, recruitment of Cdc24, but not Bem1, induces a metastable pool of Cdc42 that is sustained by positive feedback. Upon Cdk1 activation, recruitment of either Cdc24 or Bem1 creates a stable site of polarization that induces budding and inhibits formation of competing sites. Local perturbations have therefore revealed unexpected features of polarity establishment.

*For correspondence: mglotzer@uchicago.edu

**Competing interests:** The authors declare that no competing interests exist.

## Introduction

Cell polarization occurs in all kingdoms of life. In metazoa, it is critical for many cellular events including cell migration, embryogenesis, and cytokinesis. Polarization is dynamic, enabling cells to reorient to changing spatial and temporal cues. The central, conserved regulator of cell polarity in eukaryotes is the small Rho-family GTPase, Cdc42 (*Johnson and Pringle, 1990*; *Etienne-Manneville, 2004*; *Macara, 2004*). Establishment of an axis of polarity frequently involves the accumulation of active Cdc42 at a unique position on the cell cortex. An initial cue induces Cdc42-GTP accumulation at a unique site, where it then concentrates via an amplification system. This amplification system maintains the polarity site (*Thompson, 2013*). While this general scheme of polarity establishment is fairly well established, the dynamic molecular interactions required for establishing, maintaining, and enforcing a single axis of polarity are not well understood.

One of the best studied cell polarization systems is bud site selection in *Saccharomyces cerevisiae*. Wild type yeast cells use landmark-directed cues to define the polarization axis (*Figure 1A*). However, cells lacking these spatial cues or the GTPase that transduces these cues, Rsr1, efficiently establish polarity in a process known as symmetry breaking. The current model of symmetry breaking suggests that stochastic accumulation of Cdc42-GTP induces a positive feedback loop mediated by a polarity complex containing the Cdc42 GEF, Cdc24, the scaffold protein, Bem1, and Cla4, which binds directly to activated Cdc42 (*Howell et al., 2012*) (*Figure 1B*). This tripartite complex is required for polarity establishment and its molecular features provide the requisite domains to result in positive feedback (*Chenevert et al., 1992*; *Holly and Blumer, 1999*; *Bose et al., 2001*; *Butty et al., 2002*). Computational models and in vivo analyses of symmetry breaking have uncovered behaviors consistent with positive feedback, including traveling waves and oscillatory accumulation of polarity proteins (*Goryachev and Pokhilko, 2008*; *Ozbudak et al., 2005*; *Howell et al.,*

**eLife digest** Living cells are not always symmetrical. Instead they are often polarized, with a distinct front and back or top and bottom. Cell polarization influences many processes, including how a cell moves and grows, and where it will divide. Breaking symmetry – in other words, making one part of a cell different from the rest – regularly involves a small protein called Cdc42, which can switch between an active and inactive form. This protein is found in a range of organisms from fungi to animals.

Budding yeast is a valuable model to study cell polarization. This single-celled fungus polarizes in order to produce a daughter cell or 'bud' that emerges out of one end of the mother cell. To become polarized, the mother cell accumulates active Cdc42 in a small area of the cell membrane. This region then becomes the front of the cell, from where the future bud will emerge. However, it is not fully understood how active Cdc42 accumulates at only one place. One model proposed that some molecules of active Cdc42 that are already present on the membrane, recruit polarity proteins that in turn activate other, inactive Cdc42 molecules. This self-amplifying loop could eventually build up a local pool of active Cdc42. However, it has proved challenging to directly test this model.

Optogenetics is a technique in which a beam of light is used to manipulate proteins inside cells in a precise manner. The method was first developed in the field of neuroscience over a decade ago, and has more recently been applied to cell biology. Now, Witte et al. have used optogenetics to move polarity proteins to defined places on the membrane of yeast cells and analyse how this change affected the pattern of Cdc42 activation. The experiments showed that polarity proteins not only activate Cdc42 but they also recruit more polarity proteins to the same place. The resulting positive feedback loop leads to active Cdc42 accumulating at one site on the membrane. Further work showed that this mechanism only operates in this manner just before the mother cell replicates its DNA, which is when a yeast cell will normally polarize.

These results provide a new perspective on how cells can make one part of the cell different from the rest. Beyond yeast, cell polarization plays a major role when animals, including humans, are developing as embryos or healing wounds. These processes are often controlled by a protein that is equivalent to Cdc42 or by other highly related switch-like proteins. This means that yeast will continue to provide a useful model to study these processes in the laboratory. Lastly, the optogenetics approach developed Witte et al. will be useful to dissect other processes that involve molecules being located at specific place in a cell at specific time.

*2012*). However, important assumptions of the models have not been directly tested. For instance, whenever Cdc42 is active it should induce accumulation of the polarity complex; however, the validity of this critical assumption has yet to be established. Similarly, although polarity establishment has been dissected in considerable detail, the events preceding these have attracted relatively little attention. For example, in symmetry breaking polarization, do events in early G1 influence the site of symmetry breaking? To what extent are the functions of the polarity proteins influenced by the state of the cell cycle? These omissions are largely due to the absence of tools that permit controlled perturbation of Cdc42 activation.

In recent years, a number of genetically-encoded tools have been developed that allow control of diverse proteins using light (*Tischer and Weiner, 2014*). We have developed a set of small, highly engineered optogenetic protein tags, called TULIPs, that permit light-mediated control of protein-protein interactions (*Strickland et al., 2012*). In this system, the optically-responsive protein (LOV-pep) is tethered to the plasma membrane whereas its cognate binding partner (ePDZ) freely diffuses in the cytoplasm. Illumination induces a conformational change in LOVpep that allows it to bind ePDZ, causing the rapid (<10 s) relocation of ePDZ to the membrane. LOVpep spontaneously relaxes to the dark state with a half-time of 80 s (*Strickland et al., 2012*). By fusing a protein of interest to ePDZ, we can potently and dynamically control its spatiotemporal localization using light. In previous proof of principle experiments, we demonstrated that local recruitment of Cdc24 directs formation of a mating projection in α-factor treated cells (*Strickland et al., 2012*).

Using optogenetic tools, we have probed the mechanism for yeast cell polarization. Our results provide direct evidence that positive feedback contributes to polarity establishment once Cdk1 is activated at START. However, while the current model predicts that the polarity components invariably function together in a potent positive feedback loop, we find that they do not do so prior to Cdk1 activation and that Cdc42 activation does not invariably induce the canonical positive feedback loop. Rather, we demonstrate the existence of a second positive feedback loop involving Cdc24 that precedes the canonical loop and appears to function independently of Bem1. Finally, we show that multiple nascent sites of polarization compete, and that this competition is particularly potent upon activation of the Bem1-dependent positive feedback loop. We conclude that two alternative modes of positive feedback function in concert to promote polarity initiation and establishment, with their respective behaviors under strict cell cycle control.

## Results

### Cdc24 recruitment can activate Cdc42 in both unpolarized and polarized cells

In order to probe the mechanism of cell polarization, we developed optogenetic tools to recruit yeast polarity proteins to defined sites on the cell cortex. Specifically, we co-expressed a membrane-tethered variant of LOVpep (Mid2-GFP-LOVpep) with either Bem1-ePDZ or Cdc24-ePDZ in diploid *rsr1Δ* cells. In all conditions, the TULIPs-tagged proteins were expressed under the control of a β-estradiol inducible promoter (*Louvion et al., 1993*), and the ePDZ-tagged polarity component was expressed in addition to its endogenous counterpart. We perturbed cells by illuminating at a single site with a diffraction-limited laser. The response to this perturbation was examined by following the recruitment of a Cdc42 biosensor derived from the Cdc42 effector protein Gic2 (*Tong et al., 2007*) and phase optics to observe bud emergence.

We first examined the ability of Cdc24 and Bem1 to bias the site of polarization in unpolarized cells as a function of illumination frequency. We measured the angle, θ, between the site of illumination and the position of the nascent bud. These values were linearly scaled such that budding at the center of the laser target was assigned a score of 1 and budding opposite the target was assigned a score of −1; these scores were averaged for a population of cells (Polarization Efficiency = average (1-2θ/π)) (*Figure 1D*). Recruitment of Cdc24 or Bem1 recruitment was able to bias the bud site at very low light doses; Bem1 was slightly more efficient than Cdc24 (*Figure 1D*). Additionally, recruitment of either component induced robust accumulation of active Cdc42 at the site of illumination, without altering the timing of polarization (~95 min between bud emergence events, regardless of photo-activation state or molecule recruited; data not shown). As the frequency of light increased, the ability of Cdc24 to bias the bud site remained roughly constant until the highest light dose, while the polarization efficiency of Bem1 dropped by ~50% once illumination increased to greater than 3 pulses per minute (*Figure 1F*, *Figure 1—figure supplement 1*). The reason for this drop is unclear though a similar drop is observed with ePDZ-mCherry (*Figure 1—figure supplement 2*); this may result from light-induced rupture of the LOV2-flavin mononucleotide adduct (*Kennis et al., 2004*). While symmetry breaking is predicted to be random relative to the previous bud site, the position of the new bud was not completely random. Additionally, targets were not randomly positioned, as the area around the previous bud site was underrepresented. These biases would cause a slight underestimation of polarization efficiency (*Figure 1—figure supplement 3*). Optogenetically recruited Cdc24 and Bem1 were also able to induce polarization of heterozygous *RSR1/rsr1Δ* diploids (*Figure 1—figure supplement 4*).

We next assessed how the response varied as a function of both cell cycle position and the frequency with which cells were illuminated. At all doses of light, cells that were at the start of the cell cycle (~10 min before bud emergence) activated Cdc42 in response to both Cdc24 and Bem1 recruitment. Some non-illuminated cells also exhibited Cdc42 activation at the target site, as the bud site occasionally coincided with the target position (*Figure 1D,F*). To determine whether cells were constitutively responsive to optogenetic recruitment of Cdc24 or Bem1, we illuminated polarized cells with small to medium-sized buds. At infrequent light pulses, polarized cells did not activate Cdc42 in response to Cdc24 or Bem1 recruitment (1–3 pulses, *Figure 1E,F*, *Figure 1—figure supplement 5*). When cells were illuminated at a greater frequency (>2 x per minute), those that

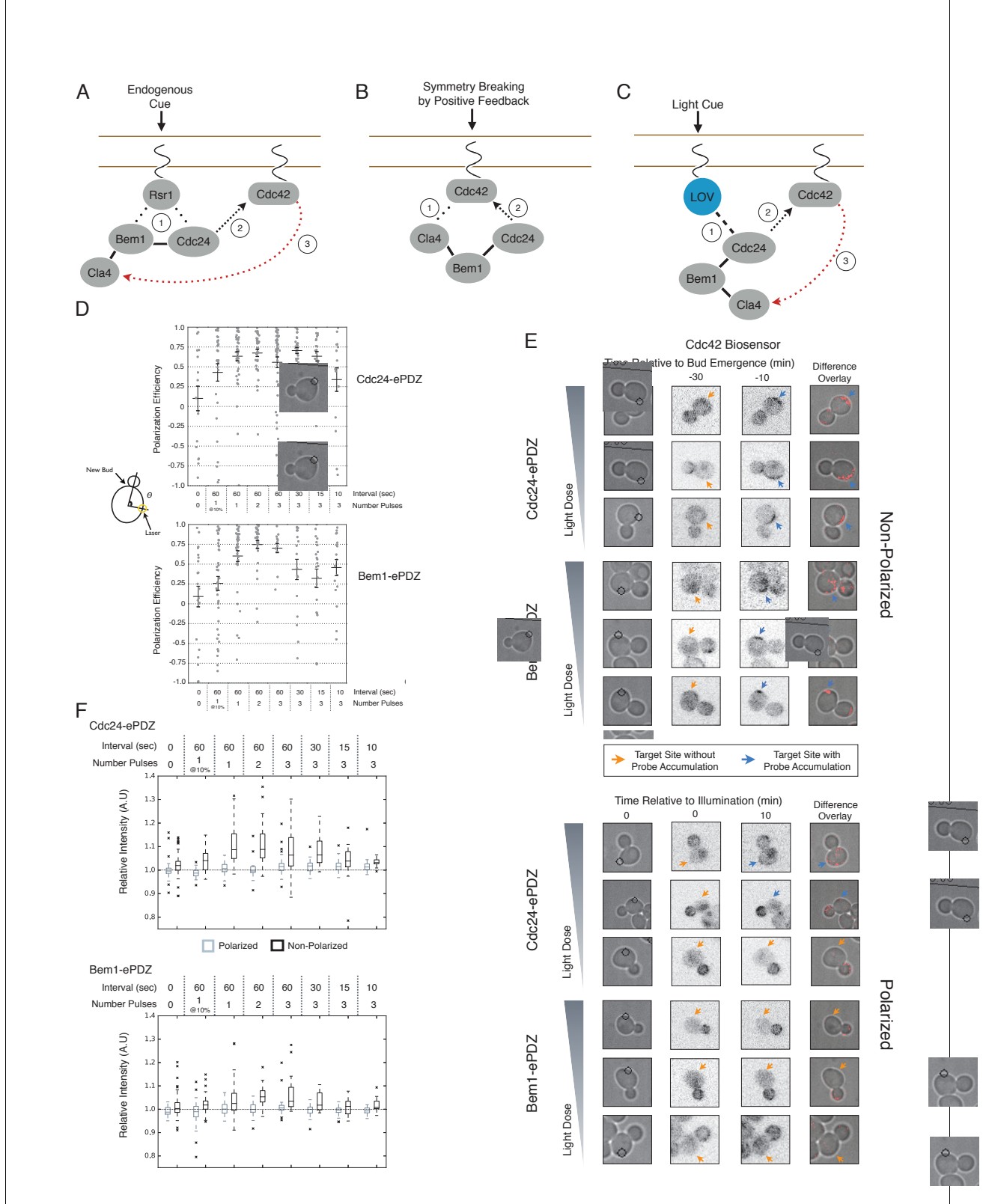

**Figure 1.** Cdc24 recruitment can induce Cdc42 activation in polarized and unpolarized cells. (**A**) The endogenous cue is mediated by a system involving Rsr1 to yield patterned budding. Rsr1 directly interacts with Bem1 and Cdc24 to recruit and activate Cdc42 at an adjacent bud position. Rsr1 recruits Bem1 and/or Cdc24 (1) to activate Cdc42 (2) adjacent to the previous bud neck. Cdc42 undergoes positive feedback (3) by interacting with Cla4 to promote its own accumulation. (**B**) In the absence of Rsr1, cells undergo a symmetry breaking event mediated by positive feedback. The symmetry

*Figure 1 continued on next page*

*Figure 1 continued*

breaking event may involve a stochastic accumulation of Cdc42-GTP at a unique location that then recruits the Cla4-Bem1-Cdc24 complex (1). Cdc24 activates additional molecules of Cdc42 (2) to promote positive feedback (3). (**C**) Light-induced symmetry breaking by recruiting the GEF Cdc24 to activate Cdc42 at a prescribed position and induce positive feedback. Localized photo-activation of LOVpep recruits Cdc24-ePDZb (1) to activate Cdc42 (2). Activated Cdc42 interacts with Cla4 to induce positive feedback. (**D**) Polarization efficiency of a population of cells where each point represents an individual cell. The angle Θ is defined by the angle between the site of bud emergence and the laser position. Data are averages of all cells across multiple experiments (n experiments >= 2, N total cells > 15 for each group). Average and ± SEM is indicated. Statistical analysis in *Figure 1—figure supplement 1*. Strains used: WYK8440 and WYK8301. (**E**) Representative phase and inverted fluorescent images depicting the activation of Cdc42 in response to either Cdc24 or Bem1 recruitment in polarized or non-polarized cells. Difference overlay images place the subtraction of the two fluorescent images overlaid onto the phase contrast image. An increase in fluorescent signal is depicted by red pseudo-coloring on the overlay. Cells were treated to increasing doses of light. Shown from bottom to top, representative images for each group:1 pulse/60 s, 3 pulses/ 30 s, and 3 pulses/15 s. Each image is 16.2 μm x 16.2 μm. Strains used: WYK8440 and WYK8301. (**F**) Box-and-Whisker plots depicting the relative change in mean fluorescence intensity of the Cdc42 biosensor at targeted regions. The relative intensity of polarized cells is the mean of the intensity at 10 min post initial illumination normalized to a control site (typically 180° from target) on the same time. The relative intensity for non-polarized cells compares accumulation of the biosensor 10 min before bud emergence normalized to a control site at the same time. Data is combined across multiple experiments (n experiments >= 2, N total cells > 15 for each group). Statistical analysis in *Figure 1—figure supplement 5*. Strains used: WYK8440 and WYK8301.

The following figure supplements are available for figure 1:

**Figure supplement 1.** Statistical analysis of polarization efficiency as a function of light dose.

**Figure supplement 2.** Recruitment of ePDZ-mCherry as a function of light dose.

**Figure supplement 3.** Bias in target position and new bud position relative to the previous bud.

**Figure supplement 4.** Local accumulation of either Cdc24 or Bem1 is sufficient to override the landmark-directed pathway.

**Figure supplement 5.** Statistical analysis of Cdc42 biosensor accumulation in polarized and non-polarized cells as a function of light dose.

expressed Cdc24-ePDZ activated detectable Cdc42 at all stages of the cell cycle. Conversely, Bem1 recruitment did not result in Cdc42 activation in polarized cells, even at higher illumination frequencies (*Figure 1E,F*, *Figure 1—figure supplement 5*). In summary, Cdc42 activation in unpolarized cells is readily induced in response to either Cdc24 or Bem1 recruitment; however, only high levels of Cdc24 recruitment can generate Cdc42-GTP in polarized cells, indicating that limitations to Cdc42 activation exist in polarized cells.

From these data, we conclude that local recruitment of either Cdc24 or Bem1 is able to efficiently bias the bud site. Because our primary interests lie in the endogenous regulation of Cdc42 activation, we chose to dissect polarity establishment using a light dose (3 pulses per minute) that efficiently biases the bud site, rather than the higher doses that might overcome the mechanisms that limit Cdc42 activation to one site in the cell.

## Bem1 recruitment induces positive feedback

To understand the role of Bem1 in regulating polarization, we induced Bem1 recruitment in cells expressing either the Cdc42 biosensor, Cdc24-tdTomato expressed on a low-copy plasmid from its endogenous promoter, or cells in which one copy of Bem1 was tagged with tdTomato. Optogenetic recruitment of Bem1 was sufficient to promote activation of Cdc42, accumulation of Cdc24 (*Figure 2A,B*), and was able to efficiently position the bud site (*Figure 2D*). Critically, we found that recruitment of Bem1 was able to induce the accumulation of endogenous Bem1 at the prescribed site, directly showing that polarization in yeast proceeds via positive feedback (*Figure 2A,B*). These results raise the possibility that the observed activation of Cdc42 represents a combination of the direct activation of Cdc42 by the Cdc24 directly interacting with optogenetically-recruited Bem1 and amplification by endogenous mechanisms. Although Bem1 recruitment efficiently induced cell polarization, it did not induce precocious Cdc42 activation, or precocious accumulation of Cdc24 or Bem1 as compared to control cells (*Figure 2B*, *Figure 2—figure supplement 1*). Thus, the activity of the Cdc24-Bem1-Cla4 feedback loop appears sensitive to cell cycle regulation.

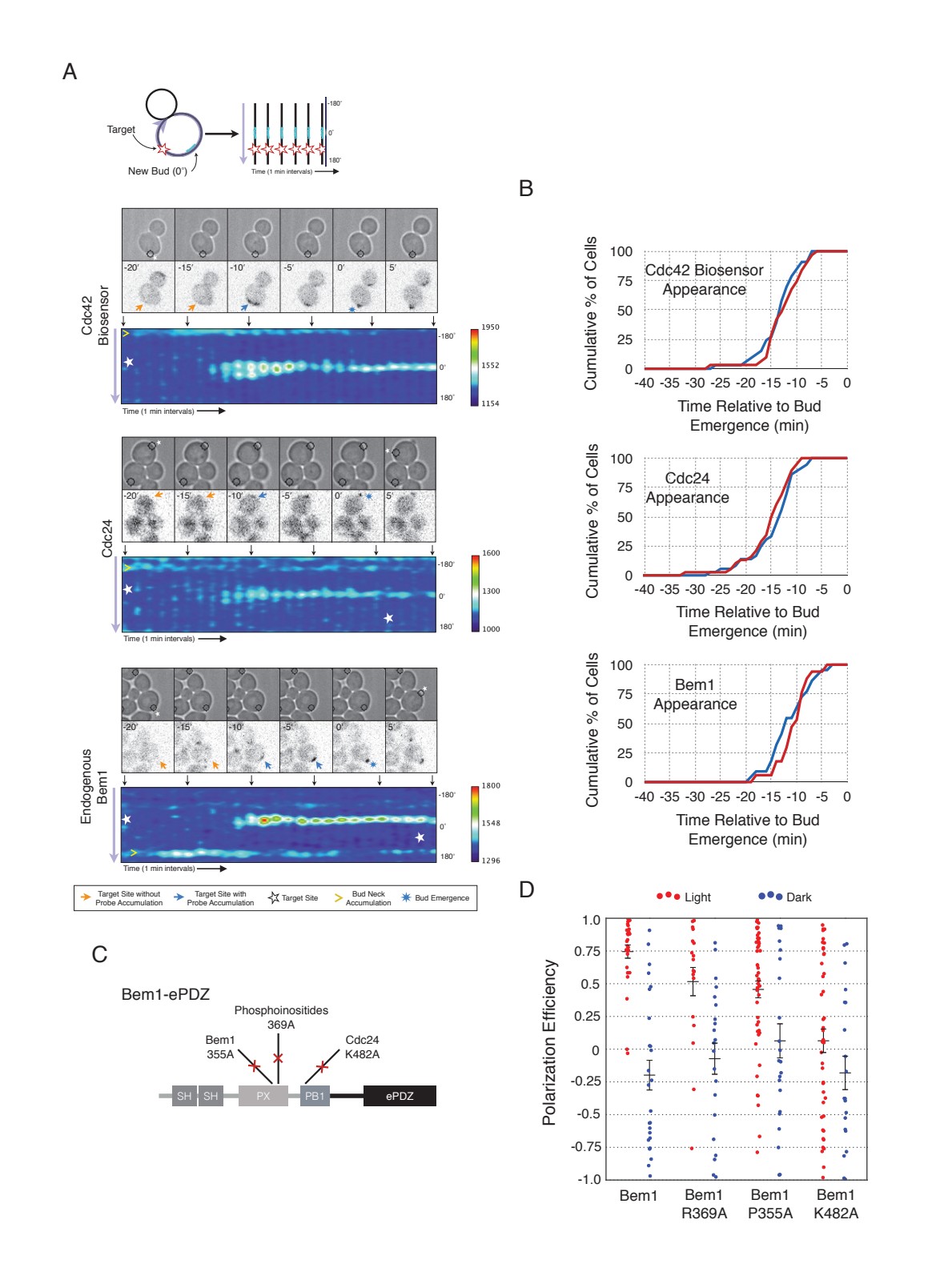

**Figure 2.** Local accumulation of Bem1 directs bud site positioning via positive feedback. (**A**) Representative phase and fluorescence images and kymographs showing the position of the laser target and accumulation of the Cdc42 biosensor, Cdc24, and endogenous Bem1 (respectively). Each image is 16.2 μm x 16.2 μm. Kymographs are generated by iteratively linearizing the membrane at each time point (schematic). Bud emergence occurs at Y = 0° and at time = 0 min. Arrows between the kymograph and fluorescent images indicate equivalent time points. Strains used: WYK8308,

*Figure 2 continued on next page*

*Figure 2 continued*

WYK8318, and WYK8576. (B) Accumulation kinetics for each component in response to Bem1 recruitment. Red line is photo-activated cells, blue line is mock-illuminated cells. Bud emergence is time = 0. Data combined across multiple experiments (n experiments >= 2, N total cells > 20 for each condition). (C) Domain schematic of Bem1-ePDZ, annotated with mutation sites and interactions. (D) Polarization efficiency of a population of cells as in 1F. Red dots are single photo-activated cells. Blue dots are single mock-illuminated cells. Data is combined across multiple experiments (n experiments > 2, N total cells > 20 for each group). Error bars indicate S.E.M. Polarization efficiency of Bem1, Bem1 R369A, and Bem1 P355A in the light is statistically significant relative to Bem1 K482A light and dark, and their respective dark-state controls. Bem1 K482A was not statistically significant relative to its dark state control. p<0.05, Mann-Whitney *U* test. Strains used: WYK8308, WYK8434, WYK8435, and WYK8436.

The following source data and figure supplements are available for figure 2:

**Source data 1.**
**Figure supplement 1.** Photo-recruited Bem1 does not alter timing of accumulation of Cdc42-GTP, Cdc24 and Bem1.
**Figure supplement 2.** Recruitment of Cdc24-binding deficient Bem1 cannot induce accumulation of endogenous Bem1.

If polarization induction by Bem1 proceeds via a positive feedback loop, Bem1 likely functions by directly interacting with the GEF Cdc24 (*Kozubowski et al., 2008*). To test this prediction, we introduced a mutation in the Bem1 PB1 domain that has previously been shown to abolish the Bem1-Cdc24 interaction (K482A) (*Figure 2C*) (*Ito et al., 2001*) and tested its ability to induce cell polarization. Mutational inactivation of the Bem1 PB1 domain ablated its polarization efficiency (*Figure 2D*) and its ability to induce recruit endogenous Bem1 (*Figure 2—figure supplement 2*). These results suggest that polarity induction by Bem1 requires direct interaction with Cdc24. In addition to its Cdc24-interacting domain, Bem1 contains a PxxP motif and SH3 domain that interact, as well as a region demonstrated to interact with phosphoinositides (PI) (*Figure 2C*). We introduced mutations known to abolish these activities (*Irazoqui et al., 2003*) and assayed each for their ability to bias the bud site. Both mutants exhibited partial activity (*Figure 2D*). Although the PI-binding region is required for Bem1 function (*Irazoqui et al., 2003*), light-mediated recruitment or indirect recruitment of endogenous Bem1 might compensate for this defect in this assay. We conclude that yeast polarization involves a positive feedback loop mediated by the Bem1-GEF complex. Furthermore, these data indicate that bud site selection requires activation of Cdc42.

## Cdc24 recruitment induces precocious activation of Cdc42

To gain a mechanistic understanding of light-induced polarity establishment in response to recruitment of Cdc24 GEF, we determined which molecular features are required for induction of cell polarization (*Figure 3A,B*). In addition to GEF activity, Cdc24 also contains a PB1 domain that enables it to bind to a corresponding PB1 domain on Bem1 (*Butty et al., 2002*). Recently, it has been suggested that localization of Cdc24 is required for polarization via its ability to activate Cdc42 (*Woods et al., 2015*). As expected, we find that mutation of conserved surface residues in the GEF-GTPase interface (*Rossman et al., 2002*) rendered Cdc24 inactive for polarization activity (*Figure 3A,B*) and Cdc42 activation (data not shown). Deletion of the Bem1-binding domain of Cdc24 marginally reduced the polarization efficiency of the GEF (*Figure 3A,B*). These results indicate that local recruitment of Cdc24 polarizes yeast cell growth through its ability to activate Cdc42, and that the interaction of the recruited Cdc24 with other cellular proteins - including endogenous wild-type Cdc24 - is not sufficient to induce polarization.

To observe the dynamics of Cdc42 activation in response to optogenetic Cdc24 recruitment, we used the biosensor of active Cdc42 to visualize GTPase activation (*Figure 3C*, *Video 1*). In control cells, 50% of cells displayed Cdc42 activation 12 min prior to bud emergence (*Figure 3D*, *Figure 3—figure supplement 1*). In experiments in which Cdc24 was continuously recruited, we observed precocious Cdc42 activation; 50% of cells exhibited polarized accumulation of active Cdc42 27 min prior to bud emergence (*Figure 3C,D*). Precocious Cdc42 activation was less robust than that observed in the ~12 min prior to bud emergence. Thus, Cdc24 recruitment can induce Cdc42 activation at an earlier stage in the cell cycle than Bem1.

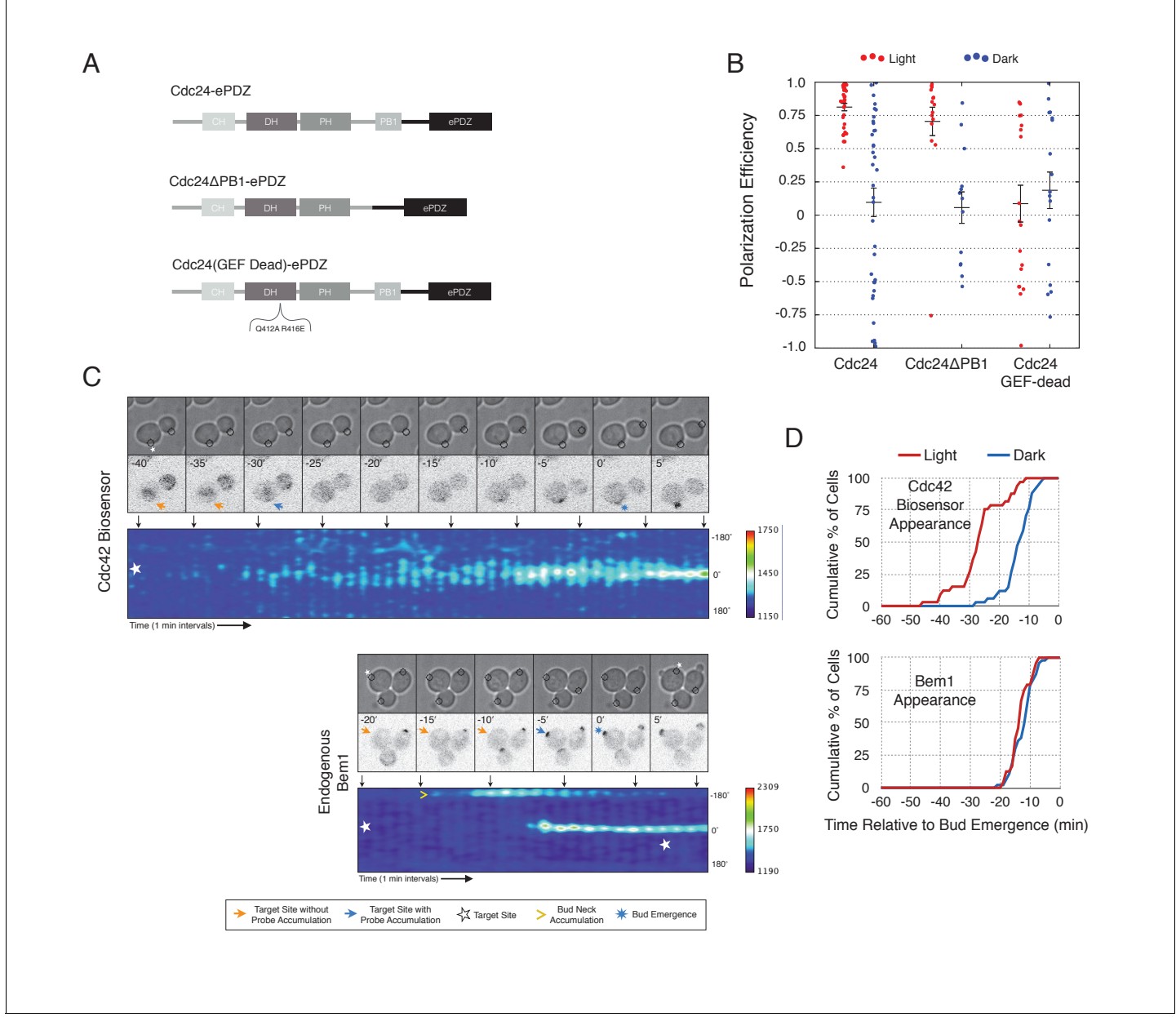

**Figure 3.** Light-mediated recruitment of Cdc24 directs bud site positioning. (**A**) Domain schematic of Cdc24-ePDZ and variants thereof. (**B**) Polarization efficiency of photo-illuminated (red) or mock-illuminated (blue) cells. Data are averaged across multiple experiments (n experiments > = 2, N total cells > 15, for each group). Average and ± S.E.M. indicated. Polarization efficiency of Cdc24 and Cdc24ΔPB1 were statistically significant relative to Cdc24-GEF-dead light/dark, and statistically significant compared to their respective dark state controls. The difference between Cdc24 and Cdc24ΔPB1 were not statistically significant. The response to Cdc24-GEF dead was not statistically significant relative to its dark state control. p<0.05, Mann-Whitney *U* test. Strains used: WYK8440, WYK8437, and WYK8439. (**C**) Panels of representative phase and fluorescence images and kymographs showing the position of the laser target and accumulation of a Cdc42 biosensor or endogenous Bem1 in response to Cdc24 recruitment. Each image is 16.2 µm x 16.2 µm. Strains used: WYK8440 and WYK8301. (**D**) Accumulation kinetics for the Cdc42 biosensor or endogenous Bem1 in response to Cdc24 recruitment. Data are combined across multiple experiments (n experiments >= 2, N total cells > 15 for each condition).

The following source data and figure supplement are available for figure 3:

**Source data 1.**

**Figure supplement 1.** Light-induced recruitment of Cdc24 induce precocious Cdc42 activation, but does not alter Bem1 kinetics.

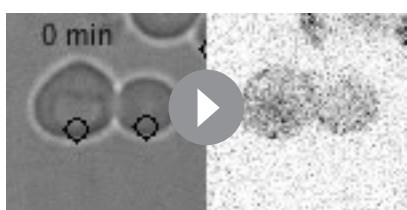

**Video 1.** Photo-recruitment of Cdc24 is sufficient to activate Cdc42 and bias the bud site. Representative phase contrast and fluorescent time-lapse images of the response to Cdc24 recruitment in cells expressing the Cdc42 biosensor. Left panel is the phase image with the position of the target defined by the black circle. Right panel is the Cdc42 biosensor. fps = 10.

We next asked whether endogenous Bem1 precociously accumulated in a similar manner as activated Cdc42. Strikingly, Bem1 did not accumulate until ~13 min prior to bud emergence (*Figure 3C,D*). Indeed, despite accumulation of active Cdc42 as a consequence of Cdc24 recruitment, Bem1 accumulates at the same time relative to bud emergence in illuminated and mock-illuminated cells (−13 min and −12 min, respectively) (*Figure 3D*, *Figure 3—figure supplement 1*). Notably, the initial accumulation of Bem1 coincides with the time in which more robust Cdc42 activation is observed. These results suggest that, unlike previous models for the activity of the polarity proteins, Cdc42 activation does not invariably lead to recruitment of the intact polarity complex.

## Cdc24, Bem1, and Cdc42 do not constitutively colocalize

The differential accumulation of Bem1 and Cdc42 in response to optogenetic recruitment of Cdc24 indicates that Cdc24 has the potential to activate Cdc42 independently of Bem1, and that activated Cdc42 does not inevitably induce recruitment of the polarity complex. If this is the case, then Cdc24, Bem1, and Cdc42-GTP may not constitutively colocalize in non-perturbed cells. To test this hypothesis, we performed a detailed colocalization analysis of the three pairwise combinations of active Cdc42, Bem1, and Cdc24. In all strains expressing fluorescent Bem1, one copy of endogenous Bem1 was tagged. The Cdc42 biosensor was used to assess the pool of active Cdc42. To visualize Cdc24, we transformed cells with a low-copy plasmid encoding an extra copy of Cdc24-GFP under its own promoter. Though it localizes to the expected sites and it can be readily detected, Cdc24-GFP was dimmer than Bem1-tdTomato, Bem1-GFP, and the Cdc42 biosensor.

Using imaging conditions optimized for detection of faint signals (see Materials and methods), we acquired maximum intensity Z-projections of asynchronous cells. Using bud size and polarization as a guide, we sub-divided the cells into unbudded G1 cells, small-budded cells, and large-budded cells and characterized the localization pattern of each pair of probes in each cell cycle state. Unbudded cells were subdivided into two groups: early and late. Early cells were characterized by small distinct puncta of all three probes. Late cells featured a wide, cortically associated band in which all three probes localized. The categorization of these patterns as early and late is substantiated by previous studies (*Ozbudak et al., 2005*) and time-lapse imaging (see below, *Figure 4C*).

As expected, all three probes colocalize at the tips of small and nascent buds. However, at other stages of the cell cycle Bem1, Cdc24, and active Cdc42 do not completely colocalize (*Figure 4A*). In large-budded cells, activated Cdc42 and Bem1 localize throughout the growing bud, while the GEF is limited to the growing bud tip. Additionally, puncta of the Cdc42 biosensor are detected in the mother cell. In early G1 cells, Bem1 and Cdc24 each localize in cortically associated puncta, but surprisingly, these puncta do not fully overlap. Furthermore, both Bem1 puncta and Cdc24 puncta displayed limited overlap with the Cdc42 biosensor, with only 35% and ~50% of puncta colocalizing, respectively (*Figure 4B*, *Figure 4—figure supplement 1*). Single plane image sequences confirm significant differences between two probes and limited differences between sequential images of a single probe. Further control experiments with co-expression of individual polarity proteins separately tagged with two fluorophores reveals extensive overlap (*Figure 4—figure supplement 2*). Notably, the localization pattern shifted in late G1 cells as all three components localize to the wide band, and the amount of reciprocal overlap between Bem1-Cdc24 and Bem1-Cdc42 biosensor increased to greater than 70% (*Figure 4B*, *Figure 4—figure supplement 1*). This more extensive overlap amongst Cdc24, the Cd42 biosensor, and Bem1 in late G1 cells suggest that prior to the onset of polarization, Cdc24 has the potential to function independently of Bem1. Of note, the punctate patterns of Bem1, Cdc24, and Cdc42 seen in early G1 cells are below the limit of detection in the assays involving optogenetic recruitment. Exposure times in those experiments were far lower in order to limit photobleaching during long-term (>90 min) imaging.

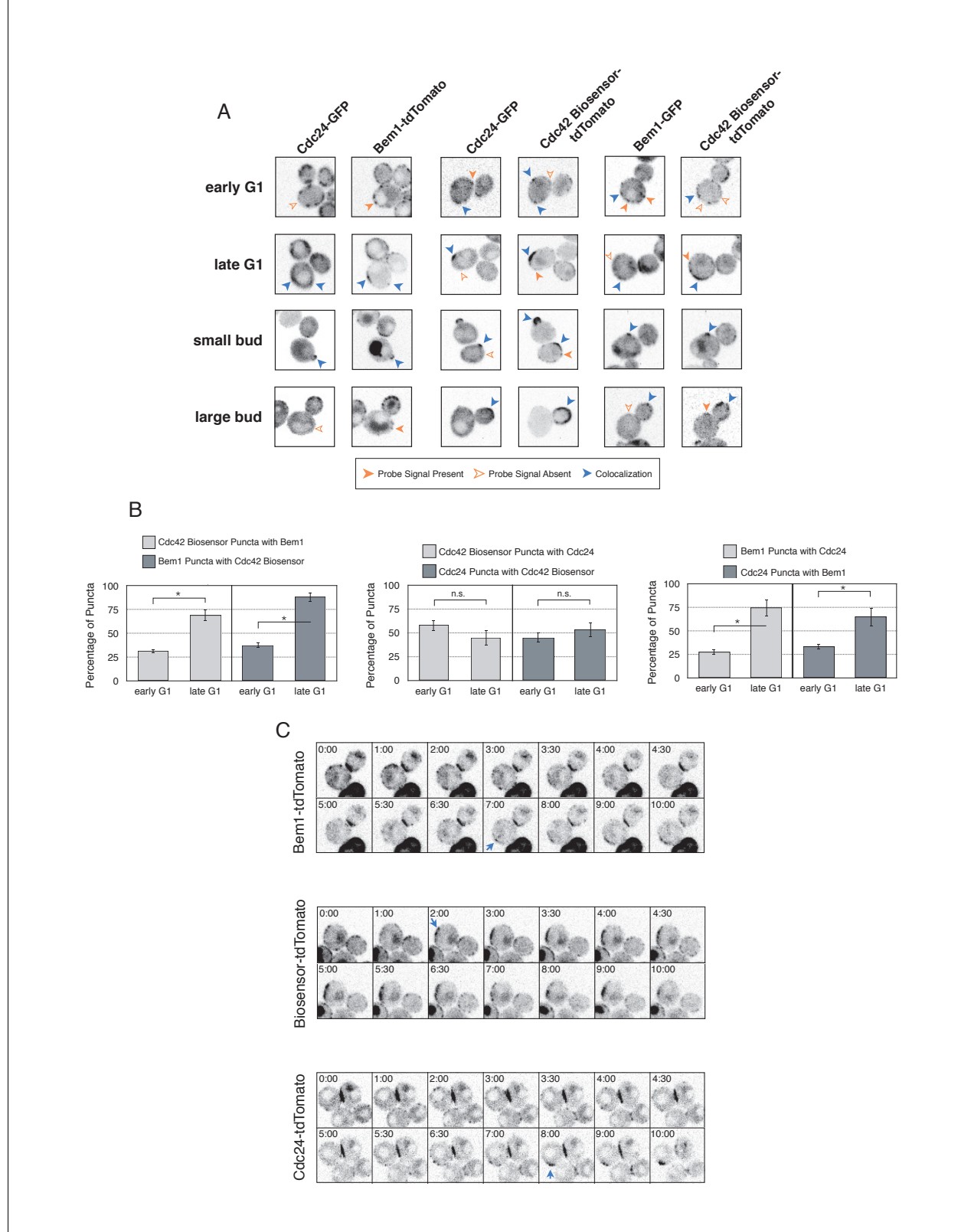

**Figure 4.** Cdc42-GTP, Cdc24, and Bem1 do not constitutively colocalize prior to polarity establishment. (**A**) Inverted fluorescent images depicting the three pairwise combinations of Cdc42-GTP, Bem1, and Cdc24. All images are Z projections of 0.25 µm slices for the center 3 µm. Each image is 13.5 µm x 13.5 µm. Strains used: WYK8550, WYK8551, and WYK8552. (**B**) Percentage of colocalization amongst puncta in early G1 or late G1 cells. Plots are separated by pairs as in **A**. Data are averages of all cells across multiple experiments (n experiments = 2; N cells > 15 for each condition; N total

*Figure 4 continued on next page*

*Figure 4 continued*

cells > 100). Error bars S.E.M. n.s indicates populations not statistically different at p>=0.05, *p<0.05, Mann-Whitney *U* test. (**C**) Time-lapse images capturing polarization in non-perturbed cells expressing either the Cdc42 Biosensor-tdTomato, endogenous Bem1-tdTomato, or Cdc24-tdTomato. Images are single planes of the center of the cell. Blue arrows denote the time and position of polarity establishment. Time = 0 is the onset of imaging, and images were captured for ten minutes at either 60 s or 30 s intervals as denoted. Strains used: WYK8301, WYK8440, and WYK8575.

The following source data and figure supplements are available for figure 4:

**Source data 1.**
**Figure supplement 1.** Pairwise analysis depicting the percent colocalization.
**Figure supplement 2.** Validation of distinct pools of polarity proteins in early G1.

To better understand the variation in protein localization at different cell cycle stages, we complemented the static imaging with time-lapse imaging. As proteins tagged with GFP tended to be dimmer than those tagged with tdTomato, we imaged cells expressing the tdTomato-tagged variants of the Cdc42 biosensor, Bem1, or Cdc24. Asynchronous populations of cells were imaged for a total of 10 min; images were acquired more frequently during the central four minutes. Each component was highly dynamic in the minutes leading up to the formation of a prominent wide band of accumulation, confirming that the punctate stage precedes polarization. While some puncta existed in the same spot for up to 3 min, appearance and disappearance of puncta were common (*Figure 4C*).

These results indicate that many Cdc24 and Bem1 molecules are not in a constitutive complex during early G1, which is consistent with the finding that recruitment of Cdc24, but not Bem1, induces Cdc42 activation at this stage of the cell cycle. Cell cycle regulated assembly of the polarity complex could readily explain both results.

## Cdk1 activation promotes Bem1 accumulation to active Cdc42

To gain insight into cell cycle regulation of the polarity complex, we performed Cdc24 recruitment studies in strains co-expressing a marker of cell cycle entry, Whi5-tdTomato, and either the Cdc42 biosensor or Bem1-tdTomato. Nuclear import of Whi5 is a marker of Cdk1 inactivation. Whi5 is concentrated in the nucleus during the interval between mitotic exit until Cdk1 activation at Start (*Costanzo et al., 2004*; *Skotheim et al., 2008*) (~25 min prior to bud emergence, *Figure 5—figure supplement 1*). Mother cells that have not imported Whi5 into the nucleus are unresponsive to Cdc24 recruitment. Conversely, Cdc24 recruitment can induce Cdc42 activation in unbudded cells with nuclear Whi5 or unbudded cells in which Whi5 nuclear export has occurred (*Figure 5A,B*), suggesting that mitotic exit is required for the activity of optogenetically recruited Cdc24. In contrast, cortical recruitment of Bem1-tdTomato in response to Cdc24 recruitment did not occur until ~12 min after Whi5 nuclear export, corresponding to ~13 min prior to budding (*Figure 5A,B*, *Figure 5—figure supplement 2*). These results are consistent with a model in which Cdk1 activation promotes assembly of the Bem1-GEF complex to engage active Cdc42.

To directly test whether Cdk1 activation is required for Bem1 recruitment in response to active Cdc42, we utilized an allele of Cdk1, cdk1-as1, that can be inhibited by an ATP-analog, 1NM-PP1 (*Bishop et al., 2000*; *McCusker et al., 2007*). Asynchronous cells were pre-treated with 1NM-PP1 for 20 min and we subsequently monitored for the ability of recruited Cdc24 to induce Cdc42 activation or Bem1 accumulation. We limited our analysis to mother cells with large-budded daughter cells, indicative of mother cells in early G1. Consistent with previous reports (*McCusker et al., 2007*), cells lacking Cdk1 activity could not undergo bud emergence nor could buds grow significantly (*Figure 5C,D*). Nevertheless, Cdc24 recruitment induced accumulation of Cdc42 in 67% of Cdk1-inhibited cells (data not shown). Indeed, Cdc42 was activated with the same kinetics irrespective of the presence or absence of the inhibitor (*Figure 5D*). Strikingly, when Cdk1 activity was suppressed, Bem1 failed to accumulate within the time frame of the experiment (greater than 1.5 hr after addition of 1NM-PP1) (*Figure 5C,D*, *Figure 5—figure supplement 3*). Furthermore, when Bem1 was recruited, cells were unable to activate Cdc42 within the time frame of the experiment (data not shown). Cells expressing wild-type *cdk1* were unaffected by the addition of 1NM-PP1

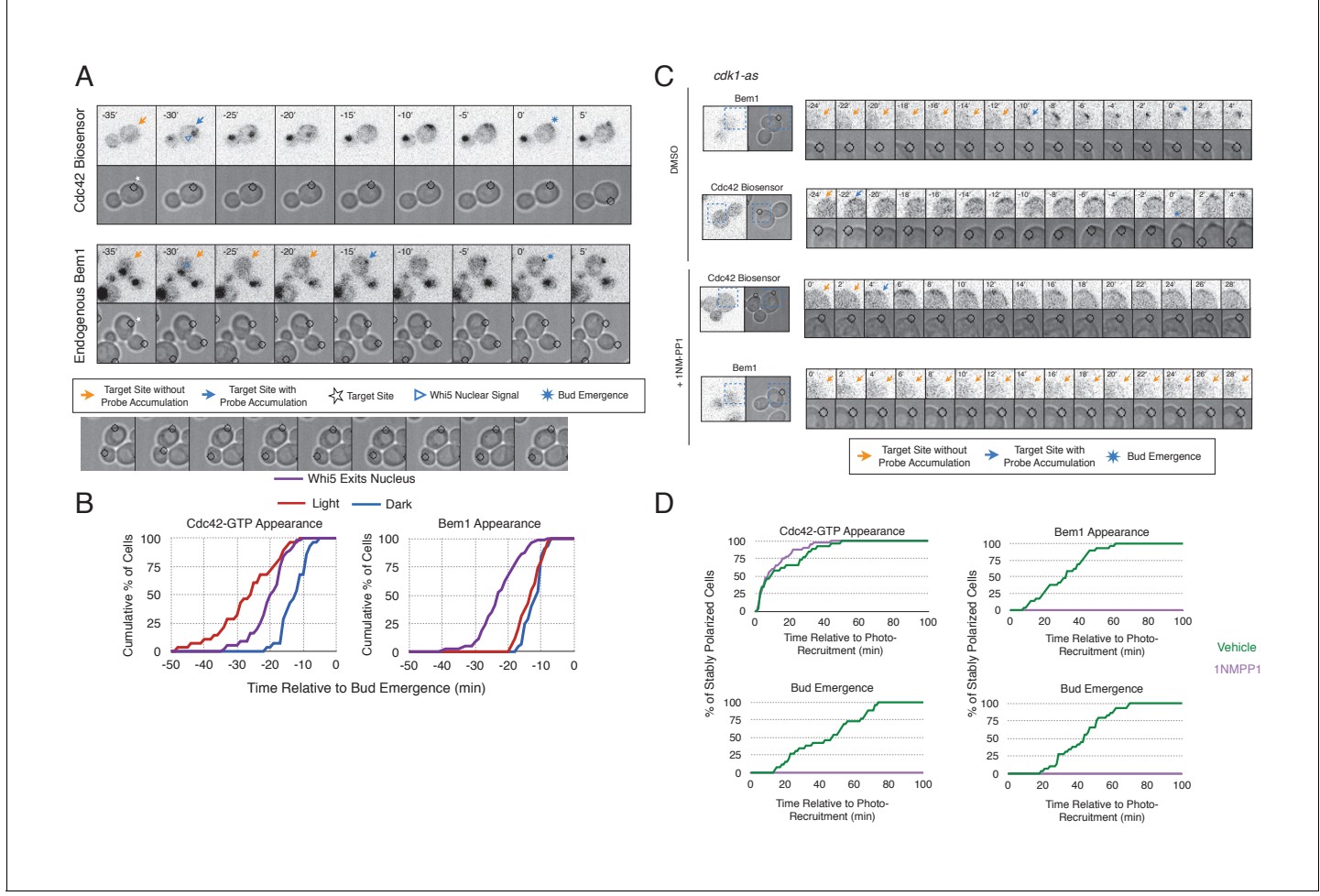

**Figure 5.** Cdk1 Activation is required for Bem1 accumulation, but dispensable for Cdc42 activation. (**A**) Representative panel of time-course images from Cdc24-ePDZ recruitment in cells co-expressing Whi5-tdTomato with either the Cdc42 biosensor or Bem1-tdTomato. Each image is 16.2 µm x 16.2 µm. Strains used: WYK8500 and WYK8502. (**B**) Whi5 nuclear exit kinetics and accumulation kinetics for either Cdc42 activation or Bem1 at Cdc24-ePDZ recruitment sites. Purple line represents Whi5 exit, with data combined for both light- and dark-state conditions as they are not significantly different (*Figure 5—figure supplement 1*). Bud emergence occurs at time = 0. Data are combined across multiple experiments (n experiments > 2; N total cells > 25 for each condition). (**C**) Representative panels and sub-images of cells depicting the response to Cdc24 recruitment +/- Cdk1 activity. Each inset is 6.5 µm x 6.5 µm. Strains used: WYK8441 and WYK8442. (**D**) Accumulation plots indicating the response of Cdc42 biosensor or Bem1 in response to Cdc24 recruitment +/- Cdk1 activity. Purple lines represent cells treated with 75 µM 1NM-PP1. Green lines represent Vehicle-treated cells. Data are combined across multiple experiments (n experiments = 2; N total cells > 25 for each condition).

The following source data and figure supplements are available for figure 5:

**Source data 1.**

**Source data 1.**

**Figure supplement 1.** Neither probe expression nor illumination affects the timing of Whi5 nuclear exit.

**Figure supplement 2.** Photo-recruited Cdc24 activates Cdc42 prior to Whi5 nuclear exit.

**Figure supplement 3.** Bem1 accumulation requires Cdk1 activity in response to light-induced recruitment of Cdc24.

**Figure supplement 4.** Addition of 1NM-PP1 to Cdk1(+) does not adversely affect Cdc42 Biosensor, Bem1 accumulation, or bud emergence.

(*Figure 5—figure supplement 4*). These results demonstrate that Cdk1 activation promotes both Bem1 accumulation and bud emergence even in cells with locally activated Cdc42.

## Cdc24 recruitment induces a Bem1-independent positive feedback loop

A prominent model for positive feedback posits that the Bem1-GEF complex is constitutive (*Bose et al., 2001*). However, our data indicates that, prior to Cdk1 activation, optogenetically-recruited Cdc24 and Bem1 function differently. The ability of Cdc24 to activate Cdc42 in the apparent absence of Bem1 lacks precedent; therefore we studied the characteristics of these optogenetically-initiated sites. Specifically, we sought to determine the stability of these sites and whether these nascent sites of polarization interact. To answer these questions, we exploited the ability of the optogenetic system to dynamically reposition the site of protein recruitment.

During the time window between the transition from isotropic growth and the time at which Cdc24 recruitment triggers Bem1 accumulation (i.e. ~13 min prior to bud emergence), Cdc24 recruitment can induce Cdc42 activation. To test whether these sites are self sustaining, the targets were removed after Cdc42 activation was detected in early G1. Optogenetic recruitment of ePDZ-tagged proteins is largely reversed within 3 min after discontinuing illumination (*Strickland et al., 2012*), ensuring that any remaining signal is not due to Cdc24-ePDZ remaining at the target site due to the optogenetic tag. Frequently, the region retained a faint but consistent signal of active Cdc42 for up to 30 min and ultimately resulted in budding from the specified site (*Figure 6A,B*). Conversely, Bem1 recruitment was only sufficient to bias the bud site if it was recruited within 15 min of bud emergence (*Figure 6A,B*). This observation shows that Cdc42 activity can be maintained at a unique site in the apparent absence of Bem1-mediated positive feedback. Thus optogenetic recruitment of Cdc24 can induce a self-sustaining pool of Cdc42 activation.

To determine how optogenetically-induced Cdc42 activation could be maintained in the absence of the optogenetic cue, we tested whether recruited Cdc24 induces a change in the localization of cytosolic Cdc24. To that end, we recruited Cdc24 and visualized Cdc24-tdTomato. Indeed, in 50% of cells in which Cdc24-ePDZ was recruited, Cdc24-tdTomato appeared on the cortex ~25 min before bud emergence, whereas in mock-illuminated cells, Cdc24-tdTomato appeared on the cortex ~15 min before bud emergence (*Figure 6C–D*, *Figure 6—figure supplement 1*). The kinetics of accumulation for Cdc24-tdTomato parallels that of activated Cdc42 in response to Cdc24 recruitment (*Figure 3*). Furthermore, catalytically inactive Cdc24-ePDZ was unable to induce Cdc24-tdTomato recruitment (*Figure 6—figure supplement 2*). From these data we conclude that recruitment of Cdc24 can induce both Cdc42 activation and Cdc24 recruitment prior to Cdk1 activation, both of which appear to occur independently of Bem1. Collectively, these results reveal the existence of a second pathway for positive feedback. The observed activation of Cdc42 likely represents a combination of direct Cdc42 activation by optogenetically recruited Cdc24 and its amplification by endogenous mechanisms.

Given that Cdc24-tdTomato accumulates at nascent sites, we hypothesized that it would remain at sites after the optogenetic cue was halted by removal of the target. Intriguingly, we observed Cdc24 was maintained at the site for ~6 min following cessation of optogenetic perturbation (*Figure 6E*). After this 6 min time window, the Cdc24 signal became more punctate and these puncta dynamically associated with the previously targeted region. Greater than 55% of cells budded from the targeted region when target removal occurred ~30 min prior to bud emergence (*Figure 6F*). These data indicate that optogenetically-initiated sites of Cdc42 activation are stable and are maintained, at least in part, by the accumulation of Cdc24, without accumulation of detectable Bem1.

## Nascent sites interact during early G1

To examine whether two sites of active Cdc42 in early G1 cells compete with one another, the target was repositioned within the same cell. Repositioning of the target caused Cdc42 accumulation at a new site and concomitant dissipation from the old site, with Cdc42 activity simultaneously detected at both sites for ~5 min (*Figure 7A*, *Video 2*). However, the site of Cdc42 accumulation could not be continuously repositioned. A qualitative change in behavior occurs prior to bud emergence. When targets are repositioned within ~13 min of bud emergence, accumulation of Cdc42 at the initial site remains and Cdc42 accumulates weakly at the new site. The cell eventually buds from the site where

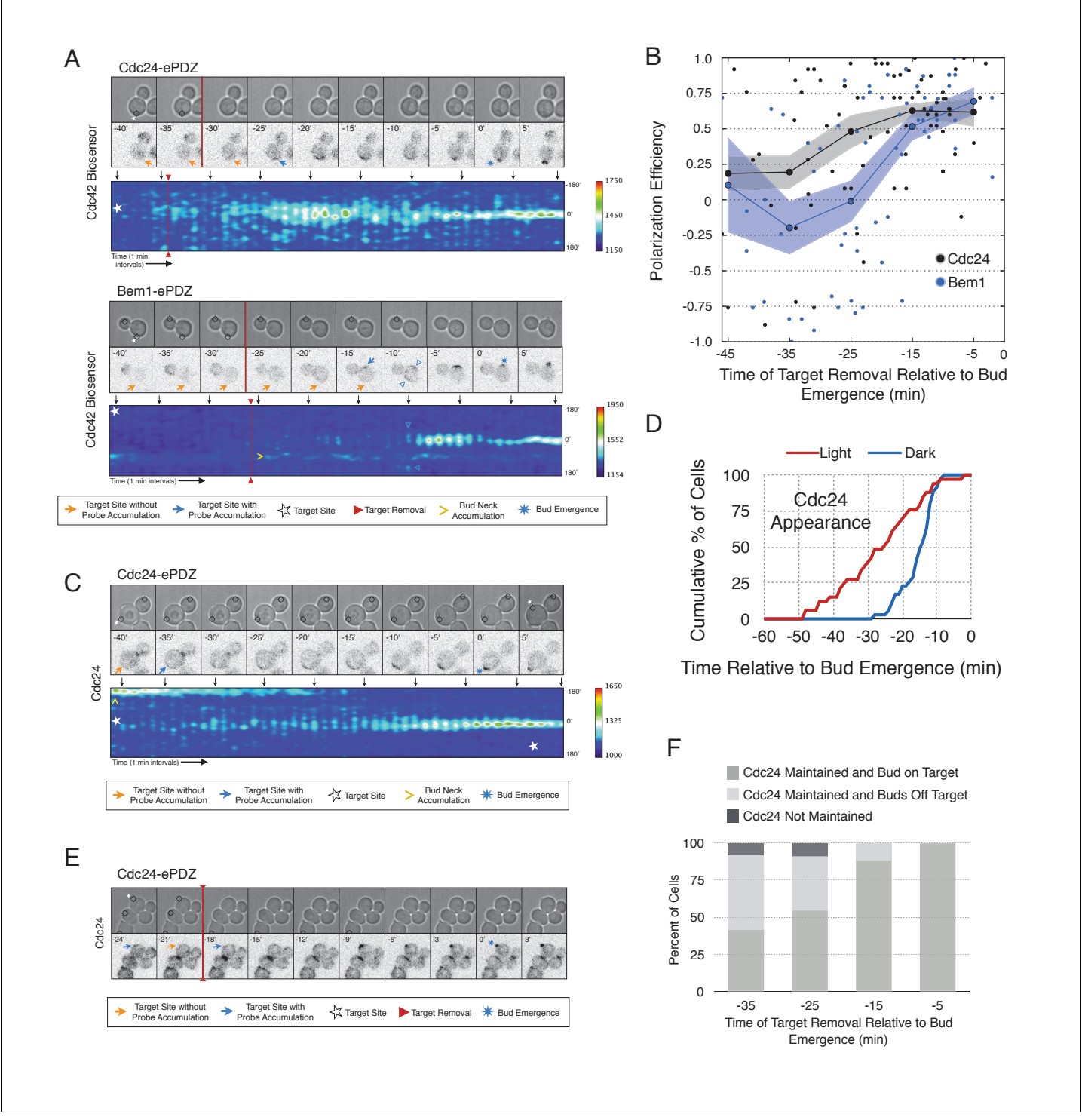

**Figure 6.** Cdc24 recruitment induces precocious Cdc42 activation and self sustaining Cdc24-tdTomato accumulation. (**A**) Phase and fluorescence images and kymographs of representative cells depicting the response to the transient recruitment of Cdc24-ePDZ or Bem1-ePDZ. Initial target site denoted by white stars; note time of target removal. Each image is 16.2 μm x 16.2 μm. Strains used: WYK8440 and WYK8308. (**B**) Polarization efficiency following transient recruitment of Cdc24 and Bem1. Time indicates when the target was removed relative to bud emergence (bud emergence occurs at time = 0). Black dots represent polarization of individual cells in response to transient Cdc24 recruitment. Blue dots represent polarization of individual cells in response to transient Bem1 recruitment. Lines represent averages (+/- SEM) of data binned in 10 min intervals, with the middle time point represented on the plot. Results are pooled across multiple experiments (n experiments >= 2; N cells > 10 for each time interval; N total cells > 75). Polarization Efficiencies of Cdc24 at −5,−15, and −25 min and Bem1 at −5 and −15 min were statistically significant relative to the corresponding

*Figure 6 continued on next page*

*Figure 6 continued*

earlier time points. Polarization Efficiencies of Cdc24 and Bem1 are statistically significant at −25 min. p<0.05, Mann-Whitney *U* test. (**C**) Representative phase and fluorescence images and kymographs showing the position of the laser target and accumulation of Cdc24-tdTomato in response to Cdc24-ePDZ recruitment. Strain used: WYK8575. (**D**) Accumulation kinetics for Cdc24-tdTomato. Data are combined across multiple experiments (n experiments > 2, N total cells > 30 for each condition). (**E**) Panels of representative phase and fluorescence images indicating the response of Cdc24-tdTomato to transient Cdc24-ePDZ recruitment. Target removal occurred at −20 min. Orange arrows denote sites of illumination without Cdc24-tdTomato accumulation. Strain used: WYK8575. (**F**) Stacked bar chart indicating the percentage of cells that maintain Cdc24-tdTomato accumulation in response to Cdc24 recruitment and whether they polarize to the prescribed site. Buds that formed within 45° of the target were defined as budding on target. Data is binned by 10 min time intervals, with the middle time point represented on the plot. Data combined across multiple experiments (n experiments > 3, N total cells > 25).

The following source data and figure supplements are available for figure 6:

**Source data 1.**
**Source data 2.**
**Figure supplement 1.** Optogenetic Cdc24 recruitment induces precocious accumulation of Cdc24-tdTomato.
**Figure supplement 2.** Recruitment of catalytically inactive Cdc24 does not induce Cdc24-tdTomato accumulation.

Cdc42 was active at the ~13 min transition point and Cdc42 signal from the alternate site dissipates upon bud emergence (*Figure 7B*). Our previous results demonstrate that Bem1 begins to accumulate at the targeted site ~13 min before bud emergence. To confirm that the basis for the qualitative switch relied on the ability of Cdc24 recruitment to induce Bem1 accumulation, we performed the same experiment in cells expressing Bem1-tdTomato. As previously shown, Bem1 only detectably accumulated at the target position defined at the ~13 min transition point and it did not accumulate at new sites if the target was repositioned prior to bud emergence (*Figure 7C,D*). These results indicate that activation of the Bem1-dependent positive-feedback loop stabilizes sites of Cdc42 activation. Furthermore, these data suggest that prior to Cdk1 activation, nascent sites of polarization influence Cdc42 activation at other sites.

Given that Bem1 accumulation at Cdc24-prescribed sites requires Cdk1 activation, we hypothesized that the dynamic properties of Cdc24-generated sites would persist in the absence of Cdk1 activation. To test this prediction, we repeated the experiment in cells expressing *cdk1-as* treated with 1NM-PP1 (*Figure 7E*). Again, limiting our analysis to mother cells with large-budded daughter cells, we found that Cdc42 activation could be dynamically repositioned for >70 min and that Bem1 was not detectably recruited within this time (*Figure 7E,F*). These results confirm that Cdk1 activation is required for Bem1-mediated positive feedback activity, which functions to establish the axis of polarity.

Combined, these results support three conclusions: (i) the ability of Cdc24 to induce Bem1 accumulation is cell cycle regulated, (ii) once active, the canonical positive feedback loop is highly stable and limits Cdc42 activation at competing sites, and (iii) maintenance of active Cdc42 and cytosolic Cdc24 before cell cycle entry appears independent of Bem1; therefore, the GTPase and the GEF participate in a positive feedback mechanism that functions earlier than the canonical Bem1-dependent positive feedback loop. This alternative positive feedback mechanism can be readily competed by a new site of Cdc24 recruitment. However, this activity may play a physiological role in establishing Cdc42 activity before Start in diploid cells.

## Actin plays a limited role in Cdc42 activation

As F-actin has been proposed to play a role in polarity establishment (*Wedlich-Soldner et al., 2003*; *Freisinger et al., 2013*; *Jose et al., 2013*), we examined whether actin depolymerization affects the response induced by local recruitment of Cdc24. Cells expressing light-recruitable Cdc24, Whi5-tdTomato and either the Cdc42 biosensor or Bem1-tdTomato were partially synchronized in G1 using a nocodazole block and release protocol and treated with Latrunculin A (LatA) to depolymerize F-actin. As previously shown, actin depolymerization partially inhibited cell polarization (*Jose et al.,*

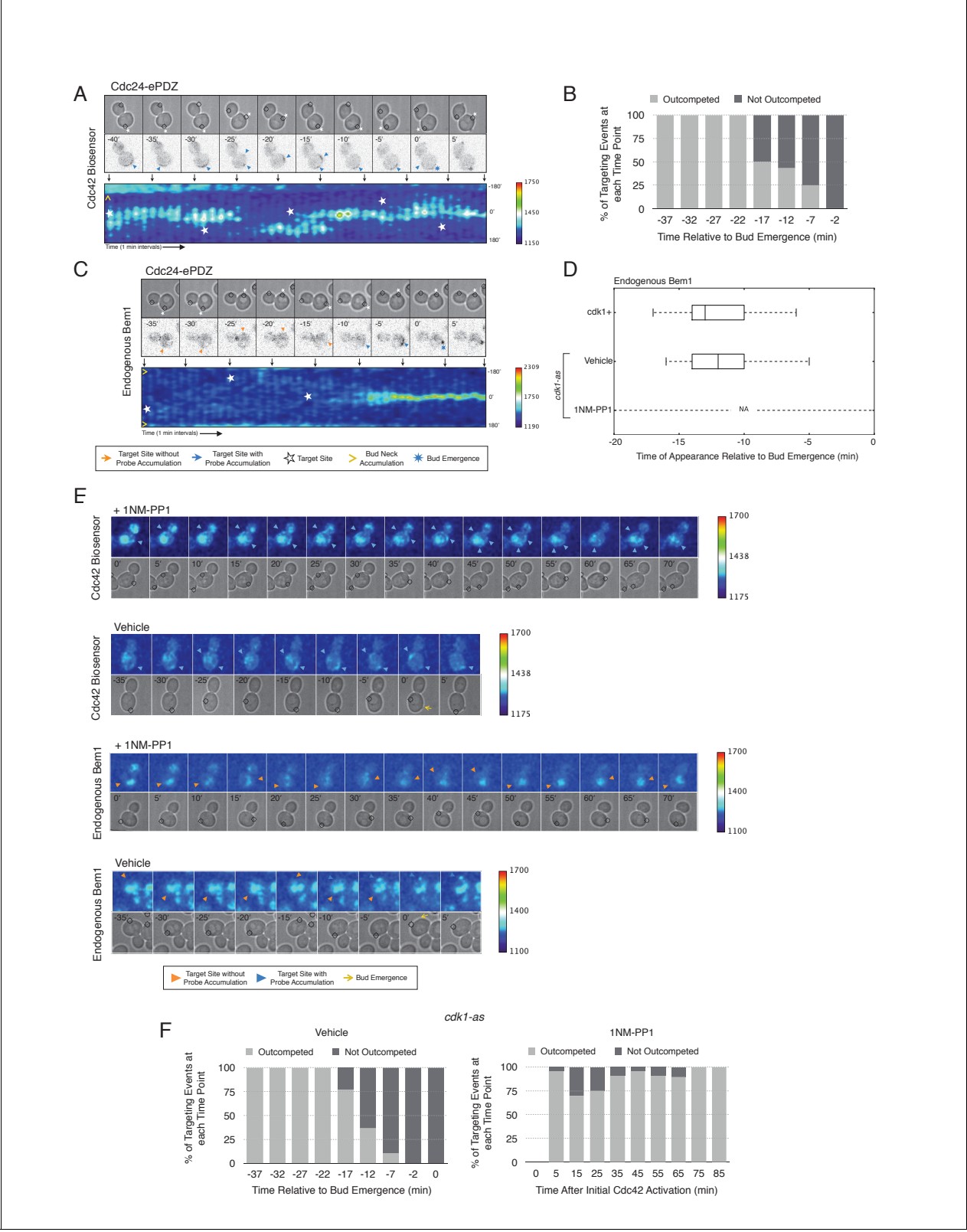

**Figure 7.** Local Cdc24 recruitment induces precocious activation of Cdc42 that is dynamically maintained in the apparent absence of Bem1. (**A**) Panels of representative cells depicting the response to dynamically re-positioned Cdc24 recruitment. Upper panels consist of phase contrast and fluorescent images and the lower panel is a kymograph. The laser was moved every 10 ± 2 min throughout the cell cycle, as denoted by the white stars. Each image is 16.2 μm x 16.2 μm. Strain used: WYK8440. (**B**) Percentage of targeting events 'Outcompeted' or 'Not Outcompeted' relative to bud

*Figure 7 continued on next page*

*Figure 7 continued*

emergence (Time = 0). Time indicates when the target was moved to a new position relative to bud emergence. Data is binned by 5 min time intervals, with the middle time point represented on the plot. The event was scored as 'outcompeted' if Cdc42 activity dissipated from the original position and accumulated at the new position. Conversely, the event was scored as 'not outcompeted' if Cdc42 activity remained at the initial position upon target repositioning. Data are combined across multiple experiments (n total experiments > 2; n targeting events per time interval >10; N total targeting events > 100; N total cells > 20). (C) Panels and a kymograph of a representative cell depicting the accumulation of Bem1 in response to dynamically repositioned Cdc24 recruitment. Strain used: WYK8301. (D) Box-and-whisker plot denoting Bem1 accumulation in CDK1, *cdk1-as* + DMSO, and *cdk1-as* + 1 NM-PP1 cells. (E) Time-course images of *cdk1-as* cells challenged with Cdc24 dynamic reorientation. Panels consist of phase contrast and either Cdc42-GTP or Bem1 accumulation pseudo-colored as a heat map. Cells were treated with either DMSO or 1NM-PP1 as indicated. Strains used: WYK8441 and WYK8442. (F) Quantification of dynamic reorientation in vehicle-treated or 1NM-PP1-treated *cdk1-as* cells. (n total experiments > 2; n targeting events per time interval >10; N total targeting events > 100; N total cells > 20) Described in B.

The following source data is available for figure 7:

**Source data 1.**
**Source data 2.**

*2013*). While 88% of cells polarize Cdc42-GTP in the presence of f-actin, only 32% of cells polarize when actin is depolymerized (*Figure 8*). Actin depolymerization has a similar effect on the efficiency of Bem1-tdTomato polarization. Drug treatment also slows cell cycle entry as judged by the efficiency of Whi5 exit from the nucleus (*Figure 8—figure supplement 1*), indicating that actin depolymerization affects several cellular processes.

Filamentous actin has been suggested to facilitate delivery of Cdc42 to the cortex where it could be activated by Cdc24 (*Wedlich-Soldner et al., 2003*). If true, Cdc24 recruitment would not be predicted to correct the polarization defect. However, light-directed recruitment of Cdc24 dramatically increased the fraction of cells that locally accumulate Cdc42-GTP in the absence of actin. Similarly, Bem1 polarization was rescued by Cdc24 recruitment in cells treated with LatA (*Figure 8*). These results suggest that actin depolymerization does not appear to affect the availability of Cdc42, but rather it impacts the localization of the Bem1-Cdc24 complex, perhaps non-specifically.

## Two sites compete to establish an axis of polarity

In the previous experiments, only one Cdc24-targeted site was active at the moment when Cdk1 was activated resulting in activation of Bem1-mediated positive feedback loop. Cdc24 recruitment to multiple sites at this critical time might result in recruitment of Bem1 at multiple sites, leading to the formation of multiple buds. Alternatively, sites may compete with each other as they form resulting in only one site becoming fully established. Therefore, to investigate whether nascent sites interact, we recruited Cdc24 to two sites simultaneously in unpolarized cells. Both sites generated activated Cdc42 (*Figure 9A*, *Video 3*) and retained active Cdc42 until ~11 min before bud emergence. Subsequently, Cdc42 activity was limited to only one site (*Figure 9B*) and bud emergence occurred at that site. In a parallel experiment with recruited Cdc24, we monitored accumulation of Bem1 and observed that it accumulates at only one of the two sites, which invariably predicted the site of bud emergence (*Figure 9A,B*, *Video 4*). Furthermore, despite recruitment of Bem1-ePDZ to two sites simultaneously, in the overwhelming majority of cases (31/33 cells), Cdc42 activation and Bem1 accumulation occurred at one site, which ultimately defined the nascent bud (data not shown). These results indicate that multiple sites cannot coexist after activation of the

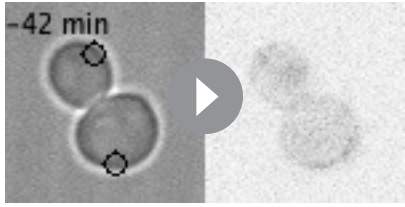

**Video 2.** Cdc42 activity can be dynamically repositioned in response to mobile sites of Cdc24 recruitment. Representative phase contrast and fluorescent time-lapse images during dynamic repositioning experiments in cells expressing the Cdc42 biosensor. Left panel is the phase image with the position of the target defined by the black circle. Right panel is the Cdc42 biosensor. fps = 10.

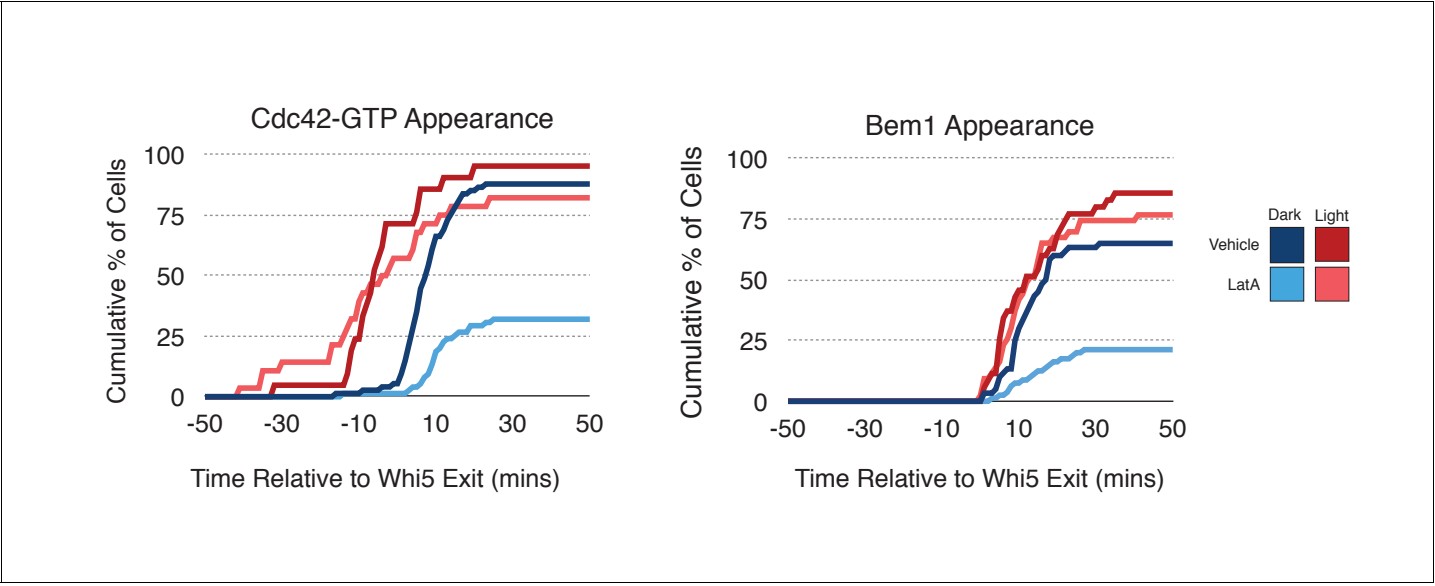

**Figure 8.** Localized recruitment of Cdc24 can overcome the polarity defect caused by actin depolymerization. Accumulation kinetics for Cdc42-GTP or Bem1 in response to Cdc24 recruitment in the presence or absence of polymerized actin. Cells were scored as 'polarized' if they retained accumulation of active Cdc42 or Bem1 for >15 min during the course of the experiment (>100 min). Red lines represent cells exposed to light-induced Cdc24 recruitment. Blue lines represent mock-illuminated cells. Dark lines represent vehicle-treated cells, while lighter-colored lines represent latrunculin A-treated cells (see schematic). Whi5 nuclear exit was used as a cell cycle marker and accumulation of Cdc42-GTP or Bem1 was scored relative to Whi5 exit. Data was binned by 5 min intervals and combined across multiple experiments (n experiments = 2; N total cells > 20). Stains used: WYK8500 and WYK8502.

The following source data and figure supplement are available for figure 8:

**Source data 1.**

**Figure supplement 1.** Loss of f-actin slows cell cycle entry.

Bem1-mediated positive feedback loop even under conditions in which they are simultaneously specified.

## Discussion

### Summary

In this work, we have studied the mechanism for Cdc42 activation throughout G1. Traditional genetic and cell biological analyses have identified the key components and revealed many behaviors associated with cell polarization, which have led to a set of models that invoke positive feedback (*Wedlich-Soldner et al., 2003*; *Irazoqui et al., 2003*; *Goryachev and Pokhilko, 2008*; *Howell et al., 2012*; *Howell and Lew, 2012*). However, the field has lacked the tools that can directly interrogate the spatio-temporal dynamics of signaling molecules and critically test the models. Here, we used optogenetics to probe the endogenous regulatory mechanisms that control Cdc42 activity. We used low light doses to recruit limited amounts of Cdc24 to reproducibly induce cell polarization; these conditions only resulted in Cdc42 activation at specific cell cycle stages. Rather than examining the direct consequences of experimentally induced GTPase activation (*Wagner and Glotzer, 2016*), we sought to generate small perturbations and define the conditions in which those perturbations were amplified by the endogenous pathways. Using this approach, we have demonstrated the existence of a previously postulated positive feedback loop which promotes bud emergence. In addition, we have demonstrated that potent mechanisms ensure that only a single site can undergo positive feedback at a given time. We have also found that the activity of this pathway is temporally confined by cell cycle regulation. Finally, we have demonstrated the existence

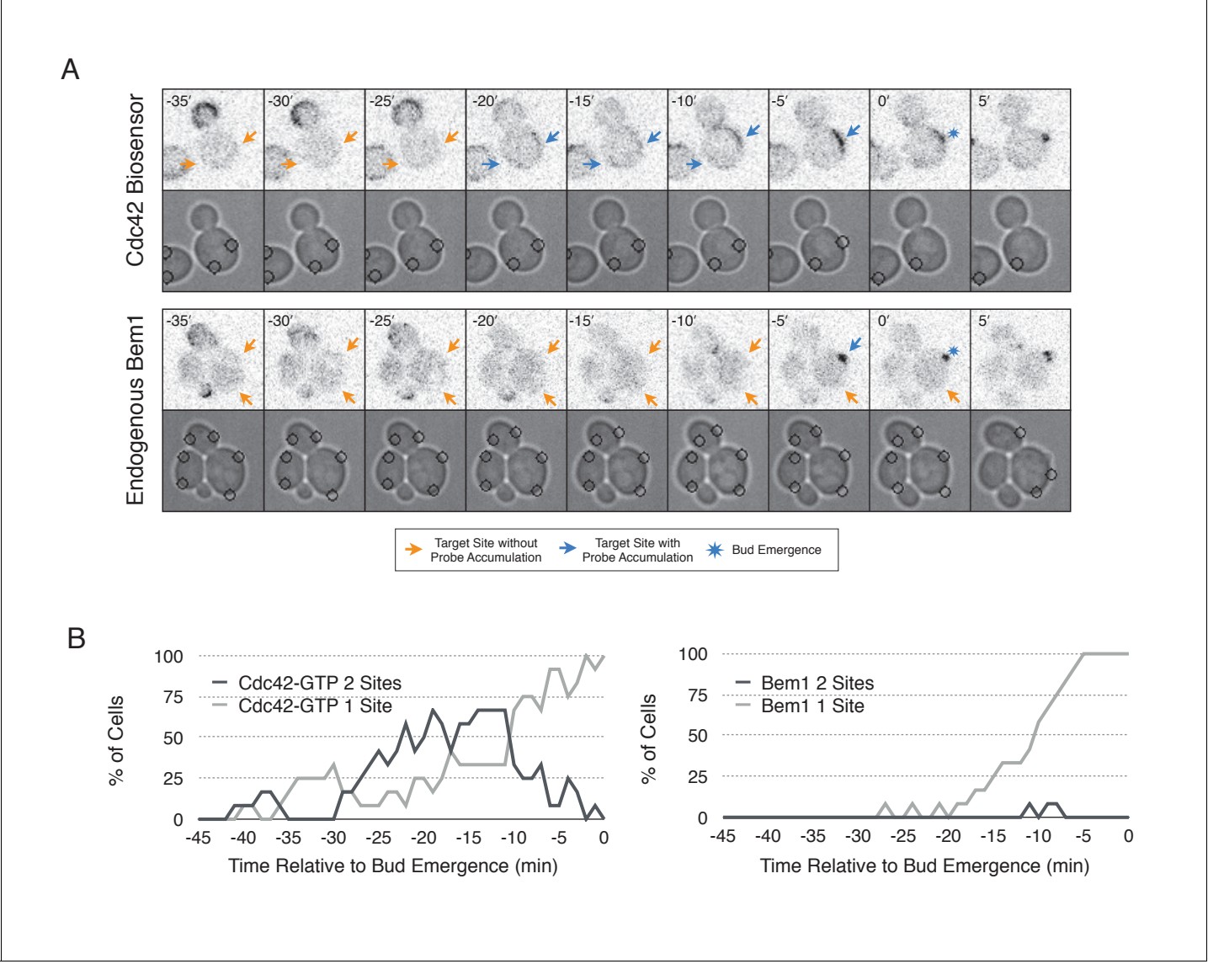

**Figure 9.** Nascent sites undergo competition to establish a single axis of polarity. (**A**) Representative fluorescence and phase images in response to simultaneous recruitment of Cdc24 to two sites. Top panel depicts Cdc42-GTP response. Bottom panel depicts Bem1 response. Each image is 16.2 μm x 16.2 μm. Strains used: WYK8440 and WYK8301. (**B**) Percentage of cells with signal at one or both sites at any given time relative to bud emergence. Dark gray lines depict percentage of cells with activation at two sites simultaneously. Light gray lines represent percentage of cells with accumulation at only a single site. Bud emergence occurs at time = 0. Data are combined across multiple experiments (n experiments >= 2; N total cells > 20 for each condition).

The following source data is available for figure 9:

**Source data 1.**

of a second, previously uncharacterized, positive feedback mechanism that can maintain a focus of Cdc42 activity prior to Cdk1 activation and which may play a role in symmetry breaking polarization.

## Positive feedback in yeast polarity

The current model of symmetry breaking polarization posits that a stochastic accumulation of Cdc42-GTP activates a positive feedback loop mediated by the Cla4-Bem1-Cdc24 polarity complex, thereby generating a local focus of activated Cdc42 (*Howell and Lew, 2012*). A prediction of that

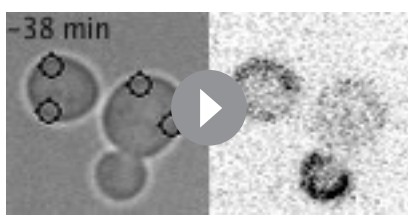

**Video 3.** Cdc42 activation can occur at two sites simultaneously in early G1 Representative phase contrast and fluorescent time-lapse images in cells expressing the Cdc42 biosensor when challenged with two sites of Cdc24 recruitment. Left panel is the phase image with the position of the target defined by the black circle. Right panel is the Cdc42 biosensor. fps = 10.

model is that a seed of Cdc42-GTP would be sufficient to define the nascent bud site. We find that local, light-induced recruitment of either the GEF Cdc24 or the scaffold Bem1 is sufficient to bias the presumptive bud site (*Figure 1*). Furthermore, the ability of these proteins to induce polarization requires the molecular features necessary to generate active Cdc42; specifically, GEF activity for Cdc24 and Cdc24-binding ability for Bem1 (*Figure 3* and *Figure 2*, respectively). These findings are consistent with the existing model for symmetry breaking, in which a positive feedback loop modulated by the polarity complex reinforces local activation of Cdc42 to promote bud emergence. We present direct evidence for positive feedback by optogenetically recruiting Bem1 and showing that endogenous Bem1 accumulates in response to that

perturbation. Likewise, Cdc24 recruitment generates positive feedback by recruiting additional molecules of Cdc24 (*Figure 6*). In both cases, the ePDZ-tagged proteins must be capable of promoting Cdc42 activation to induce accumulation of their untagged counterparts. Unexpectedly, we find that the GEF Cdc24 functions in two distinct positive feedback pathways; the Bem1-dependent loop that is active upon Cdk1 activation, and an earlier, positive feedback mechanism that appears to be Bem1 independent. The molecular mechanism of positive feedback during early G1 remains to be characterized.

Other models for symmetry breaking polarization invoke actin-dependent delivery of Cdc42 to the cortex (*Wedlich-Soldner et al., 2003*; *Freisinger et al., 2013*; *Jose et al., 2013*). While depolymerization of actin does induce a polarity defect, that defect can be readily suppressed by light-mediated recruitment of Cdc24 to the plasma membrane (*Figure 8*). This result indicates that in the absence of polymerized actin, Cdc42 still associates with the plasma membrane. Thus, actin filaments are more likely to promote, either directly or indirectly, the localization of the Cla4-Bem1-Cdc24 complex in a subset of cells.

## Cell cycle regulation of Bem1/Cdc24 complex assembly

Although recruitment of either Bem1 or Cdc24 efficiently biases the site of polarization, at certain stages of the cell cycle the molecular consequences of their recruitment are quite distinct. In late G1, following commitment to the cell cycle, light-induced recruitment of either Bem1 or Cdc24 promotes accumulation of active Cdc42, endogenous Bem1, and cytosolic Cdc24 (*Figure 2* and *Figures 3* and *6*, respectively). These results confirm prior models for polarization during the interval between Start and bud emergence. However, Bem1 recruitment does not induce precocious Cdc42 activation, Cdc24 accumulation, or endogenous Bem1 accumulation. In contrast, optogenetic recruitment of the GEF Cdc24 induces precocious activation of Cdc42 and accumulation of cytosolic Cdc24, though neither are sufficient to induce accumulation of endogenous Bem1 prior to Cdk1 activation (*Figures 3* and *5*, respectively). Synthetic lethality precluded generation of a conditional Bem1 allele compatible with our experimental approach which would be necessary to directly test the Bem1 independence of Cdc24-mediated activation of Cdc42 during early G1. Collectively, these results indicate that Cdk1 regulates the Bem1-Cdc24 complex; and, as a consequence of this regulation,

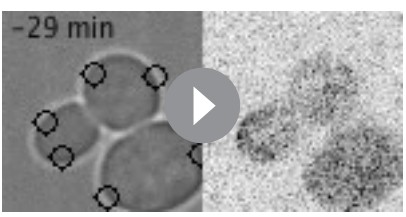

**Video 4.** Bem1 accumulation is limited to a single site of Cdc24 recruitment. Representative phase contrast and fluorescent time-lapse images in cells expressing Bem1-tdTomato when challenged with two sites of Cdc24 recruitment. Left panel is the phase image with the position of the target defined by the black circle. Right panel is the Cdc42 biosensor. fps = 10.

previous models do not apply to the period prior to Start. Rather, a second mode of positive feedback operates in this interval.

Consistent with that interpretation, we find that Cdc24, Bem1, and Cdc42-GTP do not invariably colocalize, with the lowest level of colocalization occurring prior to Cdk1 activation (*Figure 4*). We detect Bem1 and Cdc24 in small, mobile clusters that partially overlap in G1 cells. The nature of these clusters requires further analysis; though, oligomerization may contribute to their formation as the Cdc24 DH domain is capable of oligomerization (*Mionnet et al., 2008*) and Bem1 contains a PxxP motif and a SH3 domain which interact (*Irazoqui et al., 2003*; *Endo et al., 2003*) and could mediate intermolecular association. The delay in Bem1-dependent positive feedback relative to Whi5 nuclear exit (*Figure 5*) suggests that the polarity complex is regulated by Cln1/2 (*Skotheim et al., 2008*). We propose that Cdk1 activity promotes assembly of the Bem1-Cdc24 complex, this interpretation is consistent with the accumulation of Bem1 at the bud neck in early G1 without accompanying Cdc42 activation (*Atkins et al., 2013*) (*Figure 4*), as well as a Cln2-dependent increase in exchange activity towards Cdc42 (*Howell et al., 2009*).

Cdk1 activity may regulate the association of both Cdc24 with Bem1 and Bem1 with Cla4. If the former were constitutive, then Bem1 recruitment during early G1 should also result in Cdc24 recruitment which would induce Cdc42 activation (*Figure 10*). However, Bem1 recruitment in early G1 had no detectable effect on Cdc42 activation. Likewise, if the latter was constitutive, Cdc42 activation during G1 should induce Cla4 recruitment which would be predicted to induce Bem1 recruitment. However, Cdc24 recruitment in early G1 results in Cdc42 activation but does not result in detectable Bem1 recruitment. These results suggest that the canonical polarity complex is not assembled prior to Start. G1 CDK activity is implicated in directly down regulating Rga2, a GTPase activating protein that plays a role in Cdc42 regulation (*Sopko et al., 2007*; *Knaus et al., 2007*). CDK activation at start may therefore promote Cdc42 activation by at least two parallel pathways.

There also appears to be regulation of the positive feedback pathway after bud emergence. In polarized cells, Cdc24 recruitment at high light doses - but not Bem1 recruitment - activates Cdc42, raising the possibility that the well studied Cdc24-Bem1-Cla4 complex functions as a unit for a limited fraction of the cell cycle surrounding polarization.

## Coordination between Cdc42 activation before and after start

What could be the function of the secondary positive feedback resulting in Cdc42 activity before Start? Current models suggest that Cdc42 activation initiates upon Cdk1 activation (*Howell and Lew, 2012*). According to this model, stochastic activation of Cdc42 recruits the Cla4-Bem1-Cdc24 complex which amplifies this local inhomogeneity through positive feedback. Alternatively, polarization may be initiated by local association of Cla4-Bem1-Cdc24 via any of its myriad membrane association motifs. Indeed, our results show that local recruitment of members of this complex is indeed sufficient to induce symmetry breaking polarization and override the endogenous landmark-directed pathway. However, because Cdc24 can activate Cdc42 prior to Start, and because active Cdc42 is detected prior to Start (*Figure 4*), the site of polarization may not be dictated strictly by molecular noise or stochastic encounters of the Bem1-Cla4-Cdc24 complex with the membrane. Rather, a distinct mechanism(s) may exist that seeds the cortex with activated pools of Cdc42 during early G1 which pattern the 'random' choice of bud site upon passage through Start (*Figure 10*). Indeed, Bud3, which is a validated Cdc42 GEF, induces an early wave of Cdc42 activation (*Kang et al., 2014*). Though it is expressed in both haploid and diploid cells, Bud3 only impacts bud site selection in haploid cells (*Chant et al., 1995*). The linking of two mechanisms for Cdc42 activation could allow for more rapid axis specification than could be achieved by a reaction-diffusion based mechanism alone (*Goryachev and Pokhilko, 2008*; *Kang et al., 2014*). Indeed, such linked systems have been observed and modeled in a manner that can account for the speed of polarization events such as cell migration and the fixation of a single axis following an exploratory phase capable of reorientation (*Goryachev and Leda, 2017*; *Brandman et al., 2005*; *Xiong et al., 2010*).

## Competition enforces singularity

Wild-type yeast cells rarely form multiple buds in a single cell cycle, suggesting the existence of mechanisms by which nascent bud sites compete in a winner take all competition. Recent work has shown that prior to bud emergence, cells are capable of forming multiple nascent foci. Yet, by

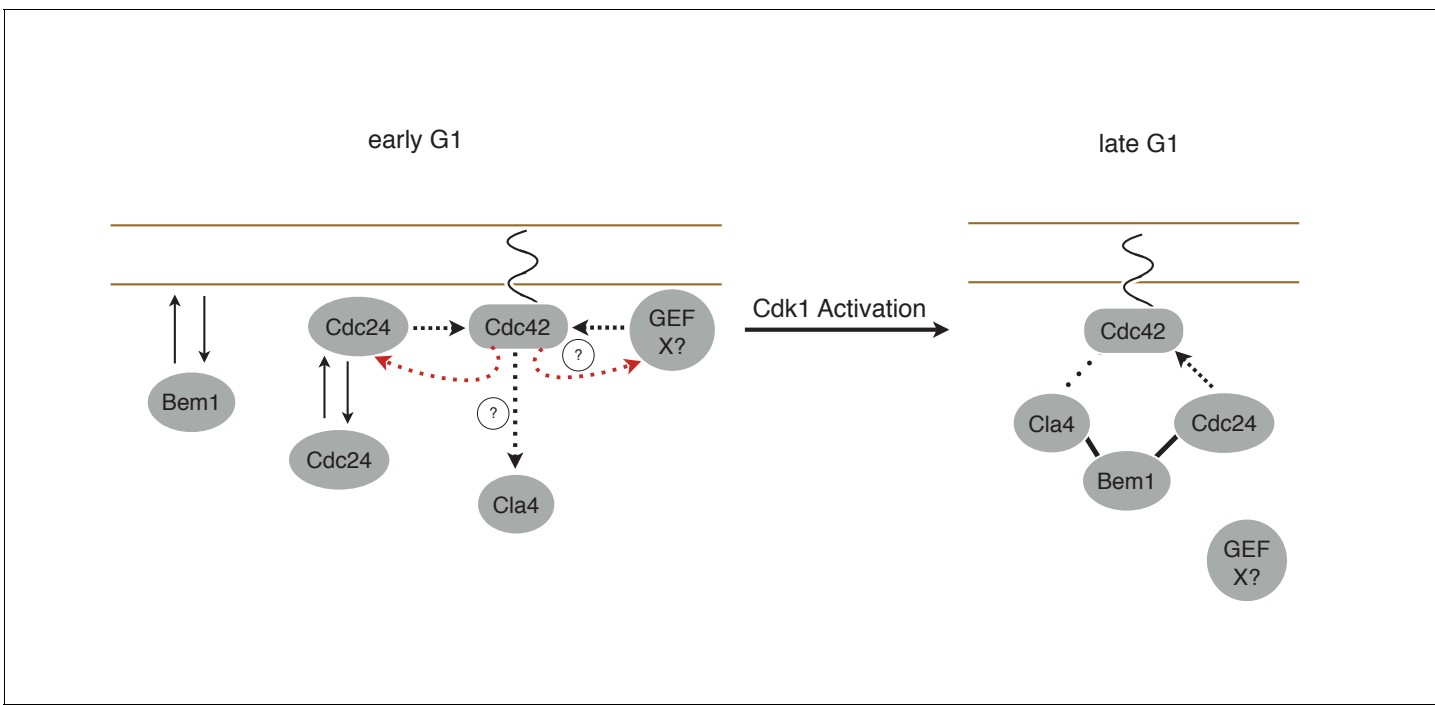

**Figure 10.** Working Model for polarity establishment. In early G1, prior to Cdk1 activation, Cdc24, Bem1, and activated Cdc42 individually associate with the plasma membrane, but do not form stable complexes. With some frequency, Cdc24, and perhaps other Cdc42 GEFs, activate Cdc42 which can participate in a weak positive feedback loop with one or both GEFs. It is unclear whether Cdc42-GTP in early G1 interact with downstream effectors, such as Cla4, but if it does, it does not initiate Bem1-dependent positive feedback. Following Cdk1 activation, the Cla4-Bem1-Cdc24 complex assembles in late G1. This complex may amplify the preexisting focus of Cdc42-GTP, and ultimately undergo strong positive feedback to generate a single focus of Cdc42-GTP that subsequently triggers bud emergence.

employing mechanism(s) that appear to involve negative feedback, only one focus matures into the site of polarization (*Howell et al., 2012*), indicating competition between the nascent foci.

Cells have been genetically manipulated to generate multiple buds. For example, expression of activated alleles of Cdc42 can induce the formation of multiple buds (*Caviston et al., 2002*). Similarly, multiple buds result from membrane-tethering either Bem1 or a Cdc24 mutant that is resistant to negative feedback, while also hindering the membrane-cytoplasm exchange of Cdc42 by deleting the facilitating chaperone, Rdi1. (*Wu et al., 2015*). However, despite the fact that transient optogenetic membrane recruitment of Bem1 or Cdc24 can induce efficient polarization, neither proved robust enough to generate two buds, indicating that the competition mechanism(s) are sufficiently strong to extinguish multiple nascent sites so that only a single axis emerges the winner. Two sites containing active Cdc42 could be generated, but we did not detect Bem1 at two sites simultaneously, consistent with models in which the intact Cla4-Bem1-Cdc24 complex is tightly limited (*Wu et al., 2015*).

Nascent sites also compete during early G1, prior to Start. Whereas a pool of Cdc42 can be induced in early G1 and maintained, if Cdc24 is recruited to a new site within the cell, the pool of active Cdc42 at the original site dissipates within a few minutes in response to the new site (*Figure 7*). As these sites appear to be Bem1 deficient, this suggests that a distinct mechanism of competition is active during this time. Indeed, whereas optogenetic recruitment of Cdc24 can induce two sites of Cdc42 activation during G1 that can co-exist, once Cdk1 is activated, the competition is more stringent and Bem1 only accumulates to detectable levels at one of the two sites.

## Optogenetics in dissection of cellular processes

In this study we show the utility of optogenetics in dissecting cellular processes. The ability to control the spatiotemporal dynamics of signaling molecules allows novel perturbations of cells that can lead to new insights. By exogenously localizing a GEF, we find that yeast cells are capable of activating

Cdc42 just after mitotic exit. We also uncovered a cell cycle regulated step in polarity establishment: activation of the canonical positive feedback loop by Cdk1 (*Figure 5* and *Figure 6*). Furthermore, we discovered a second positive feedback loop that functions independently of the canonical positive feedback mechanism and is active prior to Cdk1 activation (*Figure 6*). Finally, because optogenetics allows proteins to be dynamically repositioned in one or more spots we found that yeast cells are remarkably resistant to formation of multiple sites of polarization.

## Materials and methods

### Plasmid and strain construction

DNA manipulations were simulated with SnapGene (GSL Biotech). Plasmids were generated using a combination of conventional ligation, homologous recombination (SLICE) in bacteria (*Zhang et al., 2012*), and Gibson Assembly (*Gibson et al., 2009*). All plasmids were verified by DNA sequencing.

All strains (*Supplementary file 1A*) were constructed in the W303 background (*leu2-3 112 ura3-52 can1-100 ade2-1 his3-11 trp1-1*). Haploid a and α cells were mated by incubating overnight in 500 μL YPD and then plating on selective media. All integrating plasmids were of the YIplac series. All low copy plasmids were of the pRS series (*Supplementary file 1B*). Gene deletions were generated by one-step gene disruptions using standard procedure. Yeast were transformed using lithium acetate, single-stranded carrier DNA and polyethylene glycol. All strains were verified by colony PCR. Endogenous genes were epitope tagged by one-step PCR (*Longtine et al., 1998*). PCR products for C-terminal td-Tomato fusions were amplified from a plasmid containing a tdTomato:: HIS3MX cassette (DLB3299, a generous gift from Danny Lew). Cdc24 and Bem1 point mutants were assumed to accumulate at levels comparable to the wild-type proteins; most mutants were previously characterized (*Ito et al., 2001*; *Irazoqui et al., 2003*; *Butty et al., 2002*), the Dbs:Cdc42 co-crystal structure (*Rossman et al., 2002*) was used to identify point mutations to inactivate Cdc24 GEF activity. The endogenous Cdc24 promoter sequence consisted of 600 bp upstream of the *cdc24* locus that was amplified from genomic DNA and inserted by Gibson Assembly into Cen plasmids that encoded for either Cdc24-GFP or Cdc24-tdTomato.

The *cdc28-as1* allele was inserted by linearizing a hygromycin-resistant plasmid containing the *cdc28-as1* coding sequence with an AflII restriction enzyme site adjacent to the F88G point mutation (pKW50); the plasmid encoding the *cdc28-as1* allele was generously provided by Eric Weiss (pELW886).

### Treatment of cells for live cell imaging and drug treatment

Cells were grown in the dark at room temperature overnight in SC -His-Leu-Ura-Trp+Ade and diluted to OD600 = 0.1–0.2. For optogenetic experiments, cells were treated with 50 nM of ß-estradiol after 2 hr of growth to induce expression of the optogenetic components. After 2 hr of induction, cells were concentrated 10-fold to 20-fold in fresh media + ß-estradiol and prepped for imaging. Cells co-expressing Cdc24-tdTomato and Cdc24-ePDZ, were induced for 90 min to limit deleterious overexpression of the GEF. For experiments less than 2 hr, cells were imaged on a 2.5% agar pad soaked in minimal media + ß-estradiol + drug (where applicable) for >20 min. For experiments longer than 2 hr, cells were imaged in a CellASIC Onix microfluidic perfusion chamber (EMD Millipore Corporation) to provide continuous nutrients. For non-optogenetic experiments, cells were concentrated 10-fold to 20-fold after 2 hr of growth and imaged on 2.5% agar pad soaked in minimal media.

After 1.5 hr of induction with 50 nM ß-estradiol, *cdk1-as* cells were treated with 75 μM 1NM-PP1 or solvent (1% DMSO) at room temperature, in the dark, and without shaking for 20 min. Subsequently, cells were imaged for >90 min, for a total time in 1NM-PP1 of approximately 2 hr.

To synchronize diploid cells in early G1, exponentially growing cells were treated with 15 μg/ml nocodazole for 2 hr. Cells were induced with 50 nM ß-estradiol and treated with a second dose of nocodazole at 7.5 μg/ml. After 1 hr, cells were washed three times with fresh media and released into minimal media + 50 nM ß-estradiol for 30 min. Cells were treated with 100 μM Latrunculin A (Molecular Probes) or solvent (1% ethanol) for 20 min and then imaged for 90 min.

## Optogenetic manipulations

Cells were imaged on an Axiovert 200M microscope (Zeiss) equipped with a spinning disk confocal (CSU10, Yokogawa), a 20 mW, 561 nm laser (Cobolt), and an electron-multiplying charge-coupled device (EMCCD) camera (Cascade 512B, Photometrics) using a 63×, 1.4 numerical aperture objective (Zeiss). The microscope was controlled using MetaMorph (Molecular Devices). A 550 nm long-pass filter (Edmund Optics) was placed in the transmitted light path to avoid photoexciting the LOV domain when using phase optics. A galvanometer-steerable 440 nm dye laser (Micropoint, Photonics Instruments) for local photo excitation of Mid2-localized LOVpep. Illumination intensity was controlled by using an adjustable internal attenuator plate and an additional absorptive neutral density filter OD = 1 (ThorLabs) in the beam path.

For polarization experiments, cells were photo-excited using the Micropoint laser. The coordinates of targeted sites (x, y, t) were recorded with each photo-excitation. Following illumination, a confocal tdTomato image and phase contrast image were acquired. For control (dark state) experiments, the experiment was performed identically except that the Micropoint laser was off. Exposure times were 500 msec for tdTomato and 100 msec for phase contrast.

## Live cell imaging of unperturbed cells

For non-optogenetic experiments, cells were imaged with a Zeiss Axioimager M1 equipped with a Yokogawa CSU-X1 spinning disk unit (Solamere) and illuminated with 50 mW, 488 nm and 50 mW, 561 nm lasers (Coherent). Images were captured on a Cascade 1K electron microscope (EM) CCD camera controlled by MetaMorph (Molecular Devices).

For maximum intensity Z-projection snapshots, cells were imaged through the center 3 µm at 0.25 µm slices. The GFP and tdTomato images were acquired sequentially, followed by a phase contrast image. Maximum intensity projections were generated using Metamorph. Single plane snapshots were acquired at the mid-plane of the cell, with GFP and tdTomato images acquired sequentially. Exposure times were 900 msec for tdTomato, 900 msec for GFP, and 66 msec for phase contrast.

Single plane snapshots of a confocal tdTomato image and phase contrast image were acquired at the mid-plane of the cell. Cells were imaged at a rate of once per minute for 3 min, followed by a rate of once per 30 s for 4 min, and finally imaged at a rate of once per minute for 3 min. Exposure times were 700 msec for tdTomato and 66 msec for phase contrast.

## Image analysis

All images were analyzed in ImageJ (*Schneider et al., 2012*), with custom-written macros. To determine the angle between the laser position and the nascent bud site, we generated kymographs displaying the laser position, the site of bud emergence, and the intensity of the probe. To generate the kymographs, the cell outline was tracked using the plugin JFilament (*Smith et al., 2010*), which created a series of XY-coordinates that was converted into an ROI for each image in the stack. The ROI was then overlaid on its cognate image, linearized, and normalized to 100 pixels in length by six pixels in width. This was repeated iteratively through each slice of the stack to build kymographs. The kymographs were annotated with the center coordinate of each target in each frame of the time-lapse, the time and position of bud emergence, and the position of the previous bud site.

Polarization Efficiency was defined as $(1-2\theta/\pi)$) where $\theta$ is the angle between the site of illumination and the position of the nascent bud. A population measure of polarization efficiency was found by taking the average of the polarization efficiency value for all cells.

To quantify the appearance time of a polarity component, the time of bud emergence, and the polarization efficiency, we analyzed time-lapse images as follows: cells were only scored if they underwent both polarization and bud emergence within the time-course of the movie. The analysis was limited to mother cells; cells that budded within the first 20 min were excluded. Whi5 nuclear exit was defined as when nuclear Whi5 signal equaled that of the cytoplasm. To quantify the appearance of weak fluorescent signals, accumulation was scored by blinding the fluorescent image and scoring probe accumulation manually, by eye.

To analyze colocalization, cells were separated into cell cycle stages depending on both the bud size and the distribution of the probes. Data were blinded and cells were pulled at random from

each cell cycle subset. Puncta with colocalization (GFP with tdTomato and tdTomato with GP) were manually counted.

For *cdk1-as* cells, we limited our analysis to large-budded mother cells that accumulated Bem1 or the Cdc42 biosensor to the bud neck, indicative of early G1 cells. A 'large-bud' was considered to have an area more than 4.5 µm2. As *cdk1-as* cells treated with 1NM-PP1 do not undergo bud emergence, to be scored as polarized, they needed to maintain polarization for >15 min.

To quantify competition in dynamic reorientation experiments, a targeting event was scored as 'Outcompeted' if the Cdc42 biosensor signal disappeared from the initial position and accumulated at the new position within the time that the target was maintained at the new position. The event was deemed 'Not Outcompeted' if the Cdc42 biosensor neither disappeared from the initial position nor accumulated at the new position.

## Acknowledgements

This work was supported by NIH R01GM85087. KW was funded (in part) by NIH T32 GM007183, and D.S. was supported by an American Cancer Society Postdoctoral Fellowship (119248-PF-10-134-01-CCG). We thank Eric Weiss and Danny Lew for generous gifts of reagents and Ed Munro, Mike Rust, Dave Kovar, Danny Lew, and members of the Glotzer lab for helpful discussions and support.

## Additional information

### Funding

| Funder | Grant reference number | Author |
| --- | --- | --- |
| National Institute of General Medical Sciences | R01GM85087 | Michael Glotzer |
| American Cancer Society | 119248-PF-10-134-01-CCG | Devin Strickland |
| National Institute of General Medical Sciences | T32 GM007183 | Kristen Witte |

The funders had no role in study design, data collection and interpretation, or the decision to submit the work for publication.

### Author contributions

KW, Conceptualization, Data curation, Software, Formal analysis, Validation, Investigation, Visualization, Methodology, Writing—original draft, Writing—review and editing; DS, Conceptualization, Investigation, Methodology; MG, Conceptualization, Supervision, Funding acquisition, Visualization, Writing—original draft, Project administration, Writing—review and editing

### Author ORCIDs

Michael Glotzer, http://orcid.org/0000-0002-8723-7232

## Additional files

### Supplementary files

• Supplementary file 1. This file contains table S1 containing the list of *S. cerevisiae* strains used in this study and Table S2 containing a list of plasmids used in this study.

• Supplementary file 2. This file contains a summary of the tests for statistical significance.

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
