## [Decision Letter]

[Editors’ note: a previous version of this study was rejected after peer review, but the authors submitted for reconsideration. The first decision letter after peer review is shown below.]

Thank you for choosing to send your work entitled "Cell cycle entry triggers a switch between two modes of Cdc42 activation during yeast polarization" for consideration at *eLife*. Your full submission has been evaluated by Vivek Malhotra (Senior Editor) and three peer reviewers, one of whom, Sheila McCormick, is a member of our Board of Reviewing Editors, and the decision was reached after discussions amongst the reviewers. Based on the discussions and the individual reviews below, we regret to inform you that your work will not be considered further for publication in *eLife*. We made this decision because, in our view, it would take you more than 2-3 months to carry out the additional experiments mentioned by the reviewers. However, we strongly encourage you to re-submit, provided, you can address the reviewer comments, as all reviewers think the work is interesting.

*Reviewer #1:*

I think the topic is of interest and that they use interesting tools to address the question of how a single point of polarity is established. I have to admit that the paper was tough for me to follow, as I think it was not pitched to (or at least did not consider) readers that don't work on yeast – e.g., me – I work on a polar cell (pollen tubes) and have worked on GTPases and GEFs, but struggled. For example, it was difficult to follow the logic in the subsection “The structural requirements for Cdc24-mediated polarization” – they state that deleting the BEM1-binding site makes bud site more efficient but then their conclusion was something about recruitment. A little more "Therefore we think that this means[…]" and more links between an experimental result and their interpretation would have helped.

I liked the experiment in the last paragraph of the subsection “How do cells maintain a single axis of polarity?” (Figure 7) in which they put cdc24 at 2 sites on unpolarized cells.

Introduction, second paragraph – I wondered if the word should be absence and not presence, because as written it didn't make sense to me.

*Reviewer #2:*

In this manuscript, the authors use an optogenetic system to probe the mechanism of symmetry breaking in the budding yeast *S. cerevisiae*. Building on their previously described TULIP light-induced interaction modules, they design experiments to locally recruit the Cdc42 GEF, Cdc24, or the scaffold Bem1 at the plasma membrane. Their data indicate that there is a strong cell cycle dependence on the efficacy of Cdc42-GTP polarization, with the scaffold Bem1 not being recruited during early G1 when Cdk1 activity is low. They interpret their data as a cell cycle-dependent regulation of the well-described positive feedback, which relies on Bem1 associating both with Cdc42 effectors (which bind Cdc42-GTP) and the GEF Cdc24 (which activates Cdc42).

The topic is very interesting and the approach used is highly original and promising. Unfortunately, the paper disappoints, as it falls very short of supporting the conclusions it claims to make. I have four main points, detailed below.

General set-up of the system:

The properties of the light-inducible dimerization system are not well described in this manuscript. How efficiently and how rapidly Cdc24-ePDZ is locally recruited is not shown. The timing between laser light and Cdc42-GTP localization is not shown. The off-rates are not probed nor discussed (see below).

The kymographs are also very difficult to interpret. It is not always clear why the laser site is being moved. Also, in most of the kymographs shown, there is another signal that is repetitively seen earlier than the signal at the laser site. What is this? Is that the division site? Clarification here would greatly help the reader. As it stands, it can be interpreted as polarization at a distinct cortical site at earlier timepoint, which is highly confusing regarding the timing issues that the authors are discussing.

Timing of Bem1 recruitment:

The data shown in figures do not always support the reported timings. For instance, in Figure 2, the timings of Cdc42-GTP and Bem1 recruitments to the laser-induced sites relative to this other cytokinesis (?) signal seem very similar. This suggests that initiation of a new site may possibly be impeded by sequestration of proteins at the division site, which then only become released at cytokinesis. If this time was taken as a reference point, rather than budding, would Cdc42-GTP and Bem1 behave differently from each other?

Regarding the timing of Bem1 recruitment, in Figure 3 there seems to be some Bem1 signal recruited to the laser site at the cortex from the second image of the timelapse, before Whi5 nuclear exit, at least 35 min before bud emergence. In fact, if you compare to Figure 1 where the arrow denotes CRIB localization, the localization of Bem1 in Figure 3 second panel looks even stronger. I am thus not convinced by these images, which do not seem to correspond to the quantification shown in Figure 3.

Similarly, in Video 4, Bem1 appears as a very dynamic signal, which certainly overlaps with all of the marked sites of laser light. It is not clear to me in what way this is different from Video 3, where active Cdc42 is also quite transiently present at all the laser sites. Thus, I do not feel that the statement that Bem1 "accumulates at only one of the two sites, which invariably predicted the site of bud emergence" is an accurate description of the situation. In addition, Bem1 has been previously used as a marker of the polarity patch and shown to display transient multi-site accumulation and oscillatory behavior. This is a bit disconcerting that the authors here describe that they only see it at a single site.

Finally, timings indicated in the text do not correspond to those shown on the graphs. For instance, in Figure 2, 50% of Cdc42-GTP and Bem1 seem to happen at 12-13min, and about 28min, respectively, not 17 or 31 as indicated in the text.

CDK1-dependence:

The finding that the efficiency of symmetry breaking is regulated by Cdk1 activity is interesting. However, for the Cdk1 inhibition, the fact that asynchronous cells are used makes interpretation of results more difficult. Any cdc28-as cell treated with 1NM-PP1 to inhibit Cdk1 will be blocked in pre-metaphase if 1NM-PP1 was added when the cell was in this cell cycle stage. By focusing on large-budded mother cells, without monitoring mitosis, it is thus not clear that cells are G1.

Also, I do not understand the Bem1 signal on the kymograph of the treated cells in Figure 3. There should not be any cell division without CDK1 (unless Cdk1 inhibition happened after anaphase). And the authors claim there is no polarization. What is the signal seen?

The conclusion that Bem1 is unable to recruit Cdc24 (or be recruited by Cdc24) in early G1 due to low Cdk1 activity would be made much stronger if the authors repeat the Bem1-ePDZ recruitment experiment in cdc28-as cells treated with 1NM-PP1. If Cdk1 controls Bem1, the prediction is that Bem1 would be completely unable to activate Cdc42.

Feedback and dynamics:

At the beginning of the Discussion, the authors state that they "have been able to directly test the positive feedback model for symmetry breaking in *Saccharomyces cerevisiae*". The paper does not test any feedback: it tests the effect of Cdc42 GEF Cdc24 local recruitment, which leads to local Cdc42 activation (as would be expected) and local Bem1 recruitment (as would also be expected from the known direct binding between Cdc24 and Bem1). In neither of these recruitments is there a need to invoke a feedback control. In fact, they even show in Figure 1 that deletion of the Bem1-interaction motif on Cdc24 leads to better polarization, so clearly, this does not require feedback signaling. Similarly, and contrary to their first conclusion statement in the fourth paragraph of the subsection “Cdc42 exhibits weak positive feedback in G1 and strong positive feedback after Cdk1 activation”, the authors do not show that it is active Cdc42 that induces Bem1 accumulation. The data shown is that Cdc24 local recruitment by photo-activation leads to Bem1 recruitment. To show feedback, they would for instance need to locally recruit an active PAK kinase and show that this leads to local Cdc42 activation. Alternatively, with their current setup, they could probe whether endogenous Cdc24 is recruited to the site of Cdc24-ePDZ recruitment. If this is the case, it would suggest positive feedback regulation.

They also suggest that "activation of the Bem1-dependent positive-feedback loop stabilizes sites of Cdc42 activation". A prediction from this statement is that the Cdc24∆PB1-ePDZ construct, which does not interact with Bem1, would fail to stabilize the polarization site and that the site could be dynamically re-positioned beyond the 13-min timepoint that they indicate in the arrival of Bem1. The fact that this mutant displays more robust polarization than wildtype Cdc24 is not very encouraging of this possible outcome, but it should be tested. On a more semantic level, the word "stabilization" may induce some confusion. In the wildtype situation, Cdc42 activation and Cdc24 and Bem1 recruitment are all concomitant. This is only because here Cdc24 is artificially activating Cdc42 early that such a distinction can be made.

Regarding the proposed Bem1-independent feedback in early G1: I do not see evidence for this. When "the targets were removed" (which I take to mean that the laser light was switched off?), it is not clear at all how long Cdc24-ePDZ perdures at the recruitment site. What is the off-rate of the ePDZ-LovPep interaction in the dark? These experiments do not per se indicate absence of feedback, they can simply denote slow off-rate of the optogenetic system. The general parameters of the system need to be much better defined to be able to make any conclusion regarding site stability or competition.

*Reviewer #3:*

In this manuscript, Witte et al. use optogenetic tools to probe the regulation of Cdc42 activation in budding yeast. The authors employ a clever combination of techniques to address fundamental questions in the yeast polarity field with a focus on cell cycle inputs to Cdc42 positive feedback and the role of this circuit in competition between Cdc42-GTP foci. If the authors can perform the following experiments to bolster their key conclusions, this work definitely merits publication in *eLife*.

1) Role of Cdk1 in Cdc42 activation. One of the key conclusions of the paper is that Cdk1 regulates the timing of the Bem1 positive feedback loop, based on the observation that Cdk1 inhibition blocks the recruitment of Bem1 and impairs competition between Cdc42-GTP foci. However, a trivial explanation would be if Cdk1 activity is required to release their Cdc24-ePDZ construct from the nucleus, so a lack of Cdk1 could impair their optogenetic input instead of modulating Bem-1-based positive feedback. To address this possibility, the authors need to quantitate their optogenetic input (levels of Cdc24-ePDZ recruitment, not just timing) in the absence and presence of Cdk1 inhibition. Furthermore, I'd like to see a quantitation of the levels, not just timing, of Cdc42 generation by optogenetically recruited Cdc24 with and without Cdk1 inhibition, as there appears to be a significant inhibition of Cdc42 activation by their optogenetic input at times significantly before bud emergence, which would either indicate that Bem1 feedback is occurring at times significantly prior to bud emergence, that Cdk1 modulates Cdc24 activation directly (rather than via Bem1 positive feedback), or that Cdk1 operates through inhibiting the GAPping of Cdc42-GTP, as has previous been observed (PMID: 17853895-this reference should be cited and discussed in the revised manuscript).

2) On a related note to point 1, it is also important to quantitate Bem1-ePDZ recruitment (in Figure 4) to ensure the lack of precocious Cdc42 activation by this input isn't due to a lack of optogenetically-induced Bem1-ePDZ recruitment. This would also help clarify whether the lack of Cdc24 recruitment at earlier timepoints of optogenetic recruitment of Bem1-ePDZ could stem from Cdk1 modulation of Bem1/Cdc24 interaction.

3) Role of Bem1 in positive feedback and competition. Throughout the paper Bem1 recruitment is used as a proxy for Bem1 positive feedback. The effect of perturbations such as Cdk1 inhibition is made in this light. Because Bem1 recruitment is not observed more than 15 minutes prior to bud emergence, the authors suggest that Bem1 positive feedback is absent at earlier stages. To test whether Bem1 has a functional role versus simply correlates with feedback, the authors need to test their optogenetic system in bem1Δ cells. I understand that these are synthetic lethal with rsr1 Δ and cannot be performed in their existing background, but there are other genetic means of destroying the landmark that are compatible with bem1 Δ (PMID: 24062340), and for some experiments (competition, role of Cdk1 in Cdc42 activation), it may not be necessary to destroy the landmarks. This is absolutely essential for clarifying which of their findings (Cdk1 regulation of Cdc42, precocious optogenetic Cdc42 activation by Cdc24, competition between Cdc42 foci only near bud emergence, etc.) are indeed dependent on Bem1 positive feedback.

4) Mechanism of competition. The authors make the striking observation that roughly equivalent optogenetically-generated Cdc42 foci can compete at the level of Bem1 recruitment. This is very surprising, as a simple Bem1 limiting component model would predict equal partitioning to both optogenetically induced Cdc24 foci. To determine whether these sites might also compete for the Cdc24-ePDZ optogenetic input (which would lead to a very different interpretation for the mechanism of competition), the authors need to quantitate the recruited Cdc24-ePDZ during the competition experiments.

5) The paper lacks any statistical tests. P values should be given for all experimental conditions that are being compared. This is important, because a number of statements are made (for example, Bem1P335A mutant is more detrimental than R369A, Figure 4) that don't appear to be statistically significant.

[Editors’ note: what now follows is the decision letter after the authors submitted for further consideration.]

Thank you for resubmitting your work entitled "Cell cycle entry triggers a switch between two modes of Cdc42 activation during yeast polarization" for further consideration at *eLife*. Your article has been favorably evaluated by Vivek Malhotra (Senior Editor) and three reviewers, one of whom is a member of our Board of Reviewing Editors and one of whom is a new reviewer.

In this paper, the authors use an original optogenetic strategy to dissect the molecular mechanisms of Cdc42 polarization in budding yeast. They show that light-induced recruitment of the GEF Cdc24 or the scaffold protein Bem1 to specific cortical sites promotes local Cdc42 activation and biases the site of subsequent bud formation. This response depends on the cell cycle stage and the protein recruited. The authors conclude from their experiments that (i) they have significantly strengthened the case for a previous model of positive feedback mediated by Bem1-Cdc24 and Bem1-effector-Cdc42 interactions; (ii) they have uncovered a novel Bem1-independent positive feedback mechanism that acts earlier in the cell cycle; (iii) CDK1 activity regulates the association of Bem1 with its binding partners, so that Bem1-mediated positive feedback is gated by the cell cycle.

The optogenetic method is well-suited to the questions addressed, and the authors are to be commended for overcoming some major technical challenges. The manuscript is significantly improved, and the major conclusions may well be correct. However, there are several issues that remain unconvincing, and some aspects that were unclear and/or confusing, as detailed below. These should be relatively straightforward to fix, and if fixed this would be a valuable contribution to the field.

1) Is there really positive feedback pre-start?

This conclusion relies on two findings. First, optogenetic recruitment of Cdc24-ePDZ results in weak recruitment of Cdc24-tdTomato to the illuminated site (Figure 6). However, this signal could be due to Cdc24 oligomerization instead of positive feedback. If Cdc24-tdTomato recruitment were shown to be dependent on the catalytic activity of Cdc24-ePDZ (as was done in the biasing of the bud site experiment), that would greatly strengthen the case. Second, the signal persists after cessation of illumination. This seemed clear for Cdc42-GTP (Figure 6) but not obvious for Cdc24-tdTomato (Figure 6 and especially Figure 6—figure supplement 2). Without a convincing example of Cdc24 "maintained" after illumination ceases, it was not clear what the authors were quantifying with this label in Figure 6. In aggregate, these problems made the case for pre-start positive feedback unconvincing.

2) Is the Bem1-Cdc24 interaction really regulated by CDK1?

This conclusion also relies on two findings. First, optogenetic recruitment of Bem1-ePDZ in "polarized" cells did not lead to local GTP-Cdc42 accumulation, unlike optogenetic recruitment of Cdc24-ePDZ. This could indeed be due to a lack of interaction between Bem1 and Cdc24 at that time of the cell cycle. But it seems equally plausible that some level of interaction is constitutive, but not every Bem1 comes with an associated Cdc24. In that case, recruitment of Bem1 would lead to significantly less recruitment of Cdc24 than direct recruitment of Cdc24-ePDZ, leading to less Cdc42 activation. This should be considered as a potential explanation in the text. Second, Figure 4 shows that small puncta containing Bem1 or Cdc24 often fail to co-localize in doubly marked strains. However, there are technical concerns with this experiment, in part because the nature of the puncta is not clear: why and how would such aggregates form? Do they reflect what is happening with the uniformly distributed pools of the proteins? Might small protein numbers in the puncta lead to stochastically varying ratios of the proteins that are then interpreted as lack of co-localization? Might dynamic disassembly/movement of puncta cause an apparent lack of co-localization (e.g. if a spot moves between acquisition of the different color images)? One way to ameliorate this concern would be to image puncta in cells with two colors of the same probe (e.g. green Bem1 and red Bem1). Would these always co-localize?

3) Confusing quantification:

The statistical analysis provided in Figure 1—figure supplement 4 and B shows no significant difference between Cdc42 activation in non-illuminated cells and any of the illumination strategies for either Cdc24- or Bem1-ePDZ in "non-polarized" cells. On the face of it, this would appear to invalidate the authors' conclusions! Similarly, in Figure 1 "non-polarized" the unilluminated cells show a Cdc42 signal as high as in many of the treated cells. At what location is the signal being measured? The relevant negative control site would be a mock-illuminated site, but given the high Cdc42 signal, one suspects that the authors have quantified Cdc42 at the pre-bud site instead. The location of this measurement should be stated explicitly, and we recommend that the statistics be re-done with the unilluminated control quantified at a mock-illuminated site.

4) Untested assumptions:

The narrative makes assumptions that are not directly tested. Articulating and justifying each assumption (with citations or references to salient experiments, or with some appropriate rationale) would be very helpful:

(i) It is assumed that more frequent light pulses result in greater recruitment of ePDZ fusions.

(ii) It is assumed that mutant ePDZ fusions are expressed at similar levels to WT fusions.

(iii) It is assumed that illumination has no effect except to recruit ePDZ fusions.

5) Description of feedbacks:

The text should also be more precise in the description of feedbacks. The authors show data that is consistent with positive feedback, but do not directly demonstrate positive feedback: recruitment of Bem1 by Bem1 is consistent with feedback, but equally consistent with oligomerization, and as highlighted above, Cdc24 recruitment by Cdc24 could also be due to oligomerization. The text should be clear on these alternative possibilities. Similarly, regarding the proposed pre-start feedback, there is no direct data showing Bem1 independence. While optogenetic recruitment of Cdc24-ePDZ does not lead to detectable recruitment of endogenous Bem1, absence of evidence is not evidence of absence. Without testing that Bem1 is indeed not required, the wording of the text should be changed to make this clear.

---

## [Author Response]

[Editors’ note: the author responses to the first round of peer review follow.]

*Reviewer #1:*

*I think the topic is of interest and that they use interesting tools to address the question of how a single point of polarity is established. I have to admit that the paper was tough for me to follow, as I think it was not pitched to (or at least did not consider) readers that don't work on yeast – e.g., me – I work on a polar cell (pollen tubes) and have worked on GTPases and GEFs, but struggled. For example, it was difficult to follow the logic in the subsection “The structural requirements for Cdc24-mediated polarization” – they state that deleting the BEM1-binding site makes bud site more efficient but then their conclusion was something about recruitment. A little more "Therefore we think that this means[…]" and more links between an experimental result and their interpretation would have helped.*

We have extensively revised the text with an eye to improving clarity.

*I liked the experiment in the last paragraph of the subsection “How do cells maintain a single axis of polarity?” (Figure 7) in which they put cdc24 at 2 sites on unpolarized cells.*

Much appreciated.

*Introduction, second paragraph – I wondered if the word should be absence and not presence, because as written it didn't make sense to me.*

Presence was the intended word. The point being that, genetically, Bem1 is not strictly essential in the presence of the landmark system, though these mutants are sick (PMID 26426479). The point of the sentence in question was that, acute sequestration has a more severe phenotype than would be expected given the genetic result. Nevertheless, this sentence has been deleted, because it interrupted the flow of the paragraph.

*Reviewer #2:*

*[…] I have four main points, detailed below.*

*General set-up of the system:*

*The properties of the light-inducible dimerization system are not well described in this manuscript. How efficiently and how rapidly Cdc24-ePDZ is locally recruited is not shown. The timing between laser light and Cdc42-GTP localization is not shown. The off-rates are not probed nor discussed (see below).*

The kinetics of the TULIPs system are characterized in Strickland, 2012, therefore we did not re-investigate these parameters. In brief, once activated the LOV domain thermally reverts to the dark state with a half time of ~80 sec. Once activated, recruitment begins in seconds and during the activation window, the LOVpep-ePDZ interaction is dynamic. This information is summarized in the revised manuscript.

The parameters above were assessed with ePDZ-mCherry. To interrogate the on/off dynamics of Cdc24-ePDZ specifically, we expressed Cdc24-ePDZ-tdTomato and performed our optogenetic assays. However, the signal from Cdc24-ePDZ-tdTomato was too low to garner quantitative measurements. As a second approach, we used near-TIRF imaging to enhance the signal to noise ratio. Asynchronous populations of cells expressing either Cdc24-ePDZtdTomato or Bem1-ePDZ-tdTomato were exposed to a brief pulse of blue light (100 msec), and the response of the recruited molecule was followed. Both Cdc24-ePDZ-tdTomato and Bem1ePDZ-tdTomato rapidly relocated to the plasma membrane (<10 seconds) after illumination, regardless of cell cycle state. Consistent with the on/off rates that we reported in Strickland, 2012, both Cdc24-ePDZ-tdTomato and Bem1-ePDZ-tdTomato dissipated from the plasma membrane within 3 minutes. Additionally, we found that Cdc24-ePDZ could readily activate Cdc42 at all stages of the cell cycle, indicating that Cdc24-ePDZ is constitutively active. While these data were informative, it was difficult to directly compare the responses observed with global illumination and detection in near-TIRF to that observed with illumination of diffraction limited spots and detection with a spinning disk confocal as in the optogenetic assays.

To extend the previous studies to other stages of the cell cycle, we performed assays in which we activated cells with varying doses of light. We found that Cdc24-ePDZ was indeed capable of activating Cdc42 even in polarized cells. However, this required high light doses. Conversely, Cdc42 activation in response to Bem1-ePDZ recruitment was limited to non-polarized cells (though the near-TIRF experiments demonstrated that it is capable of recruitment at all cell cycle stages). This experiment is described in the new manuscript (Figure 1). Given that we are primarily interested in the regulation of Cdc42 activation, rather than the effects of ectopic Cdc42 activation, we used an illumination protocol that was close to the minimum necessary to promote activation during late G1.

*The kymographs are also very difficult to interpret. It is not always clear why the laser site is being moved. Also, in most of the kymographs shown, there is another signal that is repetitively seen earlier than the signal at the laser site. What is this? Is that the division site? Clarification here would greatly help the reader. As it stands, it can be interpreted as polarization at a distinct cortical site at earlier timepoint, which is highly confusing regarding the timing issues that the authors are discussing.*

To improve the interpretation of the data, we provide phase contrast and fluorescent images and correlate them to time points on the kymographs. We found that neither means of display was sufficiently clear on its own.

In the moving target experiments, Figure 7, targets are moved approximately every 10 minutes. In the target removal experiments, the targets are removed once Cdc42 activation was observed. In the majority of other experiments, targets were moved once the cell was fully polarized.

The additional signal on the kymographs comes from the bud neck. The polarity proteins are recruited to these sites for various durations during cytokinesis. We now indicate this on the kymograph.

*Timing of Bem1 recruitment:*

*The data shown in figures do not always support the reported timings. For instance, in Figure 2, the timings of Cdc42-GTP and Bem1 recruitments to the laser-induced sites relative to this other cytokinesis (?) signal seem very similar. This suggests that initiation of a new site may possibly be impeded by sequestration of proteins at the division site, which then only become released at cytokinesis. If this time was taken as a reference point, rather than budding, would Cdc42-GTP and Bem1 behave differently from each other?*

Bem1 and Cdc42-GTP display little co-localization at the bud neck, with Bem1 prominently accumulating there for ~10 minutes and then dissipating ~5 minutes prior to bud emergence. Remarkably, Bem1 displays dual localization for ~ 5 minutes, as it begins to accumulate at the nascent bud site ~10 minutes prior to bud emergence. Due to these confounding variables (lack of Cdc42-GTP accumulation and dual Bem1 localization) using cytokinesis as a timing event is not a readily accessible vantage point to compare kinetics. We favor the use of Whi5 nuclear export as it is an independent and previously validated measure of cell cycle timing.

*Regarding the timing of Bem1 recruitment, in Figure 3 there seems to be some Bem1 signal recruited to the laser site at the cortex from the second image of the timelapse, before Whi5 nuclear exit, at least 35 min before bud emergence. In fact, if you compare to Figure 1 where the arrow denotes CRIB localization, the localization of Bem1 in Figure 3 second panel looks even stronger. I am thus not convinced by these images, which do not seem to correspond to the quantification shown in Figure 3.*

We agree with the reviewer’s interpretation of the image shown in the previous example. We have reviewed these images and out of 27 cells, we found 2 in which Bem1 was detected at the cortex prior to Whi5 nuclear export. Upon further examination, the cell mentioned by the reviewer retained nuclear Whi5 signal for greater than 65 minutes indicating it is aberrant. We have therefore replaced the image with a more representative example.

*Similarly, in Video 4, Bem1 appears as a very dynamic signal, which certainly overlaps with all of the marked sites of laser light. It is not clear to me in what way this is different from Video 3, where active Cdc42 is also quite transiently present at all the laser sites. Thus, I do not feel that the statement that Bem1 "accumulates at only one of the two sites, which invariably predicted the site of bud emergence" is an accurate description of the situation. In addition, Bem1 has been previously used as a marker of the polarity patch and shown to display transient multi-site accumulation and oscillatory behavior. This is a bit disconcerting that the authors here describe that they only see it at a single site.*

We are grateful for the reviewer’s sharp observation. We have tracked back our images and realized that this video was mislabeled. In that video, we imaged the recruitment of the Cdc42 biosensor in response to Bem1 recruitment. That video did show a weak, transient activation of Cdc42 that at times overlaps with the “non-budding target”. We have reviewed the videos in which Bem1 was recruited and out of 33 cells, we found 2 in which Cdc42 was detected at both targets and 0 in which Bem1-tdTomato was detected at both targets.

*Finally, timings indicated in the text do not correspond to those shown on the graphs. For instance, in Figure 2, 50% of Cdc42-GTP and Bem1 seem to happen at 12-13min, and about 28min, respectively, not 17 or 31 as indicated in the text.*

Thank you for noticing this error, which has been corrected.

*CDK1-dependence:*

*The finding that the efficiency of symmetry breaking is regulated by Cdk1 activity is interesting. However, for the Cdk1 inhibition, the fact that asynchronous cells are used makes interpretation of results more difficult. Any cdc28-as cell treated with 1NM-PP1 to inhibit Cdk1 will be blocked in pre-metaphase if 1NM-PP1 was added when the cell was in this cell cycle stage. By focusing on large-budded mother cells, without monitoring mitosis, it is thus not clear that cells are G1.*

In these experiments, we used a nocodazole block and release protocol to enrich for cells in early G1. The fact that a parallel experiment with Cdc24 shows robust activation at low light doses provides an internal control for these experiments.

*Also, I do not understand the Bem1 signal on the kymograph of the treated cells in Figure 3. There should not be any cell division without CDK1 (unless Cdk1 inhibition happened after anaphase). And the authors claim there is no polarization. What is the signal seen?*

Relating to an earlier point, the signal was bud neck accumulation. However, the kymographs were confounding for this experiment so we now show only panels. Bud neck accumulation of Bem1 provides an internal verification that the large-budded cells are in G1, rather than premetaphase cells. This is detailed in the Materials and methods section.

*The conclusion that Bem1 is unable to recruit Cdc24 (or be recruited by Cdc24) in early G1 due to low Cdk1 activity would be made much stronger if the authors repeat the Bem1-ePDZ recruitment experiment in cdc28-as cells treated with 1NM-PP1. If Cdk1 controls Bem1, the prediction is that Bem1 would be completely unable to activate Cdc42.*

We have done this experiment and obtained the predicted result which is mentioned in the new manuscript.

*Feedback and dynamics:*

*At the beginning of the Discussion, the authors state that they "have been able to directly test the positive feedback model for symmetry breaking in Saccharomyces cerevisiae". The paper does not test any feedback: it tests the effect of Cdc42 GEF Cdc24 local recruitment, which leads to local Cdc42 activation (as would be expected) and local Bem1 recruitment (as would also be expected from the known direct binding between Cdc24 and Bem1). In neither of these recruitments is there a need to invoke a feedback control. In fact, they even show in Figure 1 that deletion of the Bem1-interaction motif on Cdc24 leads to better polarization, so clearly, this does not require feedback signaling. Similarly, and contrary to their first conclusion statement in the fourth paragraph of the subsection “Cdc42 exhibits weak positive feedback in G1 and strong positive feedback after Cdk1 activation”, the authors do not show that it is active Cdc42 that induces Bem1 accumulation. The data shown is that Cdc24 local recruitment by photo-activation leads to Bem1 recruitment. To show feedback, they would for instance need to locally recruit an active PAK kinase and show that this leads to local Cdc42 activation. Alternatively, with their current setup, they could probe whether endogenous Cdc24 is recruited to the site of Cdc24-ePDZ recruitment. If this is the case, it would suggest positive feedback regulation.*

We thank the reviewer for this important comment. In the revised manuscript, we show that Bem1-ePDZ recruitment induces accumulation of endogenous Bem1 and that recruitment of Cdc24-ePDZ induces accumulation of cytosolic Cdc24-tdTomato. Critically, we show that neither Bem1 nor Cdc24 can bias the bud site if their requisite domains for Cdc42 activation are mutationally inactivated, suggesting that Cdc42 activation is required for accumulation of either endogenous Bem1 or cytosolic Cdc24-tdTomato.

Additionally, all TULIPs components are expressed as additional copies to their endogenous counterparts. Under the conditions to which the reviewer is referring, Cdc24∆PB1-ePDZ is only functional to activate Cdc42. The active GTPase can then interact with the endogenous Cdc24Bem1-Cla4 complex via its myriad Cdc42-GTP interacting motifs.

*They also suggest that "activation of the Bem1-dependent positive-feedback loop stabilizes sites of Cdc42 activation". A prediction from this statement is that the Cdc24∆PB1-ePDZ construct, which does not interact with Bem1, would fail to stabilize the polarization site and that the site could be dynamically re-positioned beyond the 13-min timepoint that they indicate in the arrival of Bem1. The fact that this mutant displays more robust polarization than wildtype Cdc24 is not very encouraging of this possible outcome, but it should be tested. On a more semantic level, the word "stabilization" may induce some confusion. In the wildtype situation, Cdc42 activation and Cdc24 and Bem1 recruitment are all concomitant. This is only because here Cdc24 is artificially activating Cdc42 early that such a distinction can be made.*

By activating Cdc42, Cdc24∆PB1 could result in Bem1 recruitment via active Cdc42 recruiting Cla4-Bem1 complex. Furthermore, as described above, Cdc24∆PB1-ePDZ was expressed in the presence of wild-type Cdc24 thus we also expect (and now show) that Cdc42 activation will result in recruitment of a Cla4-Bem1-Cdc24 complex.

Concerning the artificial early activation of Cdc42: In the new manuscript, we have performed a detailed co-localization study in non-perturbed cells that shows that Cdc42-GTP is readily associated with the membrane in early G1. At that time, Cdc24, Bem1, and Cdc42-GTP display disparate localizations; however, after entry into the cell cycle, Cdc24, Bem1, and Cdc42-GTP strongly co-localize. These data indicate that the comment “Cdc42 activation and Cdc24 and Bem1 recruitment are all concomitant,” does not fully reflect the biological situation in *rsr1Δ* cells. Our results raise the possibility that Cdc42 activation upon Cdk1 activation may involve nascent sites of polarization.

Regarding the proposed Bem1-independent feedback in early G1: I do not see evidence for this. When "the targets were removed" (which I take to mean that the laser light was switched off?).

We actually moved the targets to sites distant from cells as cells in a given field responded asynchronously.

*It is not clear at all how long Cdc24-ePDZ perdures at the recruitment site. What is the off-rate of the ePDZ-LovPep interaction in the dark? These experiments do not per se indicate absence of feedback, they can simply denote slow off-rate of the optogenetic system. The general parameters of the system need to be much better defined to be able to make any conclusion regarding site stability or competition.*

As mentioned in response to the first point raised by this reviewer, the dissociation of the optogenetic probes is fairly rapid. We have clarified this point in the new manuscript. Finally, we show that Cdc24-ePDZ induces precocious accumulation of Cdc24-tdTomato, and that Cdc24tdTomato can remain at the initiated site for upwards of ~30 minutes after cessation of Cdc24ePDZ recruitment.

*Reviewer #3:*

*In this manuscript, Witte et al. use optogenetic tools to probe the regulation of Cdc42 activation in budding yeast. The authors employ a clever combination of techniques to address fundamental questions in the yeast polarity field with a focus on cell cycle inputs to Cdc42 positive feedback and the role of this circuit in competition between Cdc42-GTP foci. If the authors can perform the following experiments to bolster their key conclusions, this work definitely merits publication in eLife.*

We thank the reviewer for their supportive comments.

*1) Role of Cdk1 in Cdc42 activation. One of the key conclusions of the paper is that Cdk1 regulates the timing of the Bem1 positive feedback loop, based on the observation that Cdk1 inhibition blocks the recruitment of Bem1 and impairs competition between Cdc42-GTP foci. However, a trivial explanation would be if Cdk1 activity is required to release their Cdc24-ePDZ construct from the nucleus, so a lack of Cdk1 could impair their optogenetic input instead of modulating Bem-1-based positive feedback. To address this possibility, the authors need to quantitate their optogenetic input (levels of Cdc24-ePDZ recruitment, not just timing) in the absence and presence of Cdk1 inhibition. Furthermore, I'd like to see a quantitation of the levels, not just timing, of Cdc42 generation by optogenetically recruited Cdc24 with and without Cdk1 inhibition, as there appears to be a significant inhibition of Cdc42 activation by their optogenetic input at times significantly before bud emergence, which would either indicate that Bem1 feedback is occurring at times significantly prior to bud emergence, that Cdk1 modulates Cdc24 activation directly (rather than via Bem1 positive feedback), or that Cdk1 operates through inhibiting the GAPping of Cdc42-GTP, as has previous been observed (PMID: 17853895-this reference should be cited and discussed in the revised manuscript).*

The reviewer raises a justifiable concern. However, our experiments are performed in diploid cells in which Cdc24 does not translocate into the nucleus. This is demonstrated in the revised manuscript (Figure 4), in which we performed a detailed colocalization study and found that Cdc24, Bem1, and Cdc42-GTP display disparate (non-nuclear) localizations prior to Cdk1 activation. Following Cdk1 activation, Cdc24, Bem1, and Cdc42-GTP strongly colocalize. These results support the interpretation that Bem1 and Cdc24 are not complexed prior to Cdk1 activation. However, this does not eliminate additional, parallel regulatory mechanisms such as inhibition of Rga2, the GAP for Cdc42.

We agree that the activation of Cdc42 by Cdc24 prior to Cdk1 activation is weaker than that following Cdk1 activation. In interpreting these effects, it is important to keep in mind that the observed accumulation of active Cdc42 is likely due to both the optogenetically recruited GEF and GEF that accumulates indirectly due to feedback. It is not possible to deconvolve these contributions. Thus the reduced activity prior to Cdk1 activation could be due to reduced recruitment of GEF via feedback, reduced specific activity of the GEF at this stage, and differential activity of the Cdc42 GAP. Our functional data shows that prior to Cdk1 activity, GEF recruitment induces Cdc42 activation, but Bem1 recruitment is not detectable. Conversely, Bem1 recruitment does not result in Cdc42 activation at these stages.

*2) On a related note to point 1, it is also important to quantitate Bem1-ePDZ recruitment (in Figure 4) to ensure the lack of precocious Cdc42 activation by this input isn't due to a lack of optogenetically-induced Bem1-ePDZ recruitment. This would also help clarify whether the lack of Cdc24 recruitment at earlier timepoints of optogenetic recruitment of Bem1-ePDZ could stem from Cdk1 modulation of Bem1/Cdc24 interaction.*

Please refer to earlier comments (reviewer 2, point 1) discussing our near-TIRF and dosage experiments. Both Bem1-ePDZ-tdTomato and Cdc24-ePDZ-tdTomato are readily recruited to the membrane upon global illumination in near-TIRF at various stages of the cell cycle. Because spot recruitment of Bem1-ePDZ-tdTomato does not result in detectable levels of Bem1-ePDZtdTomato outside of G1, this suggests that the phenotypes we observe result from the activity of the recruited proteins and their amplification by endogenous mechanisms.

*3) Role of Bem1 in positive feedback and competition. Throughout the paper Bem1 recruitment is used as a proxy for Bem1 positive feedback. The effect of perturbations such as Cdk1 inhibition is made in this light. Because Bem1 recruitment is not observed more than 15 minutes prior to bud emergence, the authors suggest that Bem1 positive feedback is absent at earlier stages. To test whether Bem1 has a functional role versus simply correlates with feedback, the authors need to test their optogenetic system in bem1Δ cells. I understand that these are synthetic lethal with rsr1 Δ and cannot be performed in their existing background, but there are other genetic means of destroying the landmark that are compatible with bem1 Δ (PMID: 24062340), and for some experiments (competition, role of Cdk1 in Cdc42 activation), it may not be necessary to destroy the landmarks. This is absolutely essential for clarifying which of their findings (Cdk1 regulation of Cdc42, precocious optogenetic Cdc42 activation by Cdc24, competition between Cdc42 foci only near bud emergence, etc.) are indeed dependent on Bem1 positive feedback.*

We worked diligently to generate the strains necessary for this experiment. As the reviewer mentioned, the synthetic lethality between Bem1 and Rsr1 makes this a non-trivial exercise.

Indeed, even in the presence of Rsr1, Bem1 deletion impairs cell fitness (PMID 26426479). Ultimately, we reasoned that the most controlled way to do this experiment was to conditionally inactivate Bem1 using auxin-triggered destruction. We generated a strain in which endogenous was tagged with *a 3xMini-AID:HIS3MX* cassette (PMID 23499004. When transformed with Tir4, these cells responded as expected, with growth and budding inhibited within 3 hours of auxin addition to *bem1-3xMini-AID:HIS3MX* cells (data not shown).

We next constructed and sporulated this diploid strain in the W303 background (*leu2-, trp1-, ura3-, his3-, ade2-*):

*MATa/α; bem1-3xMini-AID:HIS3MX/Bem1; rsr1*Δ:*TRP/rsr1*Δ:*KanMX; GIC2 /GIC2(1-208aa)tdTomato:HIS3MX*

We isolated 84 His+ spores which could contain GIC2-tdTomato and/or *bem1-3xMiniAID:HIS3MX*. When these strains were scored for the GIC2 marker and the Bem1-AID fusion, wefound 41 contained *bem1-3xMini-AID:HIS3MX,* and43 contained *GIC2(1-208aa)tdTomato:HIS3MX.* None of the strains contained both markers. These data indicate that *bem1-3xMini-AID:HIS3MX/GIC2(1-208aa)-tdTomato:HIS3MX* cells in an *rsr1*Δ background are synthetic lethal (even in the absence of Tir4 and auxin!).

To test whether wild-type *RSR1* would alleviate the synthetic lethality, we generated a diploid heterozygous for *RSR1/rsr1*Δ *(in the W303 background)* and performed tetrad analysis:

*MATa/α; bem1-3xMini-AID:HIS3MX/Bem1; rsr1*Δ:*TRP/RSR1; GIC2 /GIC2(1-208aa)tdTomato:HIS3MX*

We sporulated this strain and dissected tetrads, isolating 58 His+ spores. When these were scored for the GIC2 marker and the Bem1-AID fusion, we found 33 of the former, 25 of the latter, but 0 in which both alleles were present (despite being on different chromosomes). We conclude that Bem1-AID and GIC2-tdTomato are synthetically lethal (even in the absence of the TIR4 gene and auxin!).

From these data we infer that Bem1-3xMini-AID and the Cdc42 biosensor are synthetic lethal regardless of Rsr1. As the goal of the original experiment was to determine whether Cdc24 recruitment can induce activation of Cdc42 in the absence of Bem1, the synthetic lethality precludes answering that question.

*4) Mechanism of competition. The authors make the striking observation that roughly equivalent optogenetically-generated Cdc42 foci can compete at the level of Bem1 recruitment. This is very surprising, as a simple Bem1 limiting component model would predict equal partitioning to both optogenetically induced Cdc24 foci. To determine whether these sites might also compete for the Cdc24-ePDZ optogenetic input (which would lead to a very different interpretation for the mechanism of competition), the authors need to quantitate the recruited Cdc24-ePDZ during the competition experiments.*

This would be good to do, however Cdc24-ePDZ-mCherry is too dim to reliably quantify (See earlier discussion).

*5) The paper lacks any statistical tests. P values should be given for all experimental conditions that are being compared. This is important, because a number of statements are made (for example, Bem1P335A mutant is more detrimental than R369A, Figure 4) that don't appear to be statistically significant.*

We have provided an extensive statistical analysis of our results.

[Editors' note: the author responses to the re-review follow.]

*[…] The manuscript is significantly improved, and the major conclusions may well be correct. However, there are several issues that remain unconvincing, and some aspects that were unclear and/or confusing, as detailed below. These should be relatively straightforward to fix, and if fixed this would be a valuable contribution to the field.*

*1) Is there really positive feedback pre-start?*

*This conclusion relies on two findings. First, optogenetic recruitment of Cdc24-ePDZ results in weak recruitment of Cdc24-tdTomato to the illuminated site (Figure 6). However, this signal could be due to Cdc24 oligomerization instead of positive feedback. If Cdc24-tdTomato recruitment were shown to be dependent on the catalytic activity of Cdc24-ePDZ (as was done in the biasing of the bud site experiment), that would greatly strengthen the case. Second, the signal persists after cessation of illumination. This seemed clear for Cdc42-GTP (Figure 6) but not obvious for Cdc24-tdTomato (Figure 6 and especially Figure 6—figure supplement 2). Without a convincing example of Cdc24 "maintained" after illumination ceases, it was not clear what the authors were quantifying with this label in Figure 6. In aggregate, these problems made the case for pre-start positive feedback unconvincing.*

These are important points. Indeed, we have performed experiments with catalytically inactive Cdc24-ePDZ and find that recruitment of Cdc24-tdTomato does not occur – in early or late G1. If Cdc24 recruitment were simply due to oligomerization, the GEFdead variant would be predicted to be functional.

The following statement was added to the text:

“Catalytically inactive Cdc24-ePDZ was unable to induce Cdc24-tdTomato recruitment (Figure 6—figure supplement 2).”

Concerning the second point, we have provided another example that better demonstrates the maintenance in a static image. Because these foci are not perfectly stationary, the retention is frequently easier to visualize in panels with more frequent timepoints.

*2) Is the Bem1-Cdc24 interaction really regulated by CDK1?*

*This conclusion also relies on two findings. First, optogenetic recruitment of Bem1-ePDZ in "polarized" cells did not lead to local GTP-Cdc42 accumulation, unlike optogenetic recruitment of Cdc24-ePDZ. This could indeed be due to a lack of interaction between Bem1 and Cdc24 at that time of the cell cycle. But it seems equally plausible that some level of interaction is constitutive, but not every Bem1 comes with an associated Cdc24. In that case, recruitment of Bem1 would lead to significantly less recruitment of Cdc24 than direct recruitment of Cdc24-ePDZ, leading to less Cdc42 activation. This should be considered as a potential explanation in the text. Second, Figure 4 shows that small puncta containing Bem1 or Cdc24 often fail to co-localize in doubly marked strains. However, there are technical concerns with this experiment, in part because the nature of the puncta is not clear: why and how would such aggregates form? Do they reflect what is happening with the uniformly distributed pools of the proteins? Might small protein numbers in the puncta lead to stochastically varying ratios of the proteins that are then interpreted as lack of co-localization? Might dynamic disassembly/movement of puncta cause an apparent lack of co-localization (e.g. if a spot moves between acquisition of the different color images)? One way to ameliorate this concern would be to image puncta in cells with two colors of the same probe (e.g. green Bem1 and red Bem1). Would these always co-localize?*

We do not disagree that Cdk1 may regulate the fraction of Bem1 that is in complex with Cdc24, as opposed to Cdk1 being strictly required for formation of the complex. We have rephrased several statements in the Results section (related to Figure 5) from “Cdk1 is required for” to “Cdk1 activity promotes”. The language that we use in the Discussion was already quite measured in this respect, “Collectively, these results indicate that Cdk1 regulates the Bem1-Cdc24 complex”; “Cdk1 activity may regulate the association of both Cdc24 with Bem1 and Bem1 with Cla4.”

To address the technical concerns, we have performed two experiments. First, in addition to comparing the localization of the two probes in sequential images, we acquired two sequential pairs of images. For example, instead of just comparing Bem1GFP to Cdc24-tdTomato images, we acquired four sequential images: Bem1tdTomato, Cdc24-GFP, Bem1-tdTomato, Cdc24-GFP. While there was some changes in sequential images in a single channel, these changes we much more limited than those comparisons between channels (~80%; Figure 11). Note that the time interval between sequential control images in a single channel (e.g. GFP) is twice as long as the interval between channels, and therefore would tend to overestimate the difference.

To complement this control, we also examined cells in which a single species was doubly-labeled, as suggested by the reviewer. These two also showed a high degree of colocalization (~80%; Figure 11).

Author response image 1.Recruitment of Cdc24 or Bem1 that cannot activate Cdc42 do not induce accumulation of wildtype Cdc24 or Bem1, respectively.(**A**) Inverted fluorescent images depicting the pairwise combination of Bem1 and Cdc24 in early G1. All images are single planes and the number in the upper left corresponds to the order in which the images were acquired. Each image is 13.5 µm x 13.5 µm. Quantification corresponds to the percentage of puncta that remain the same, appear, or disappear between 1 and 3 or 2 and 4. Data are averages of all cells across multiple experiments (n experiments = 2; N cells > 50). Error bars S.E.M. n.s indicates populations not statistically different at p >= 0.05, * p < 0.05, Mann-Whitney *U* test. Strains used: WYK8551. (**B**) Inverted fluorescent images depicting the pairwise combination of the Cdc42 biosensor and Cdc24 in early G1. All images are single planes and the number in the upper left corresponds to the order in which the images were acquired. Each image is 13.5 µm x 13.5 µm. Quantification as in **A**. Data are averages of all cells across multiple experiments (n experiments = 2; N cells > 50). Error bars S.E.M. n.s indicates populations not statistically different at p >= 0.05, * p < 0.05, Mann-Whitney *U* test. Strains used: WYK8552. (**C**) Inverted fluorescent images depicting the pairwise combination of the Cdc42 biosensor and Bem1 in early G1. All images are single planes and the number in the upper left corresponds to the order in which the images were acquired. Each image is 13.5 µm x 13.5 µm. Quantification as in A. Data are averages of all cells across multiple experiments (n experiments = 2; N cells > 50). Error bars S.E.M. n.s indicates populations not statistically different at p >= 0.05, * p < 0.05, Mann-Whitney *U* test. Strains used: WYK8550. (**D**) Representative fluorescent images of early G1 cells co-expressing Bem1-tdTomato and Bem1-GFP. All images are Z projections of 0.25 𝜇m slices for the center 3 𝜇m. Each image is 13.5 µm x 13.5 µm. Quantification is the percentage of colocalization amongst puncta in early G1. Data are averages of all cells across multiple experiments (n experiments = 2; N cells > 75). Error bars S.E.M. n.s indicates populations not statistically different at p >= 0.05, * p < 0.05, Mann-Whitney *U* test. Strain used: WYK8554. (**E**) Representative fluorescent images of early G1 cells co-expressing Cdc24-tdTomato and Cdc24-GFP. All images and quantification as in D. Data are averages of all cells across multiple experiments (n experiments = 2; N cells > 75). Error bars S.E.M. n.s indicates populations not statistically different at p >= 0.05, * p < 0.05, Mann-Whitney *U* test. Strain used: WYK8553.**DOI:**
http://dx.doi.org/10.7554/eLife.26722.047

*3) Confusing quantification:*

*The statistical analysis provided in Figure 1—figure supplement 4 and B shows no significant difference between Cdc42 activation in non-illuminated cells and any of the illumination strategies for either Cdc24- or Bem1-ePDZ in "non-polarized" cells. On the face of it, this would appear to invalidate the authors' conclusions! Similarly, in Figure 1 "non-polarized" the unilluminated cells show a Cdc42 signal as high as in many of the treated cells. At what location is the signal being measured? The relevant negative control site would be a mock-illuminated site, but given the high Cdc42 signal, one suspects that the authors have quantified Cdc42 at the pre-bud site instead. The location of this measurement should be stated explicitly, and we recommend that the statistics be re-done with the unilluminated control quantified at a mock-illuminated site.*

[As an additional supplemental figure is now included, this comment concerns Figure 1—figure supplement 5.]

We measured targeted sites in non-illuminated unpolarized cells, as suggested by the reviewer (in other words the only difference was whether or not the photoactivation laser was on or off). A small number of cells in this population happened to polarize at the site of the “target”, essentially as suggested by the reviewer (pre-bud sites). In the case of Cdc24 this could be directly ascribed to chance, as these foci of Cdc42 appeared at the regular time ~13” prior to bud emergence, not precociously as observed when Cdc24 is recruited. The response of a small portion of cells resulted in considerable scatter in the data, in part due to the small sample size at the time. Previously we had measured the accumulation of the targeted site at 0’ preactivation and at 10’. To reduce variability we now measure accumulation at the 10’ time point at the target and at a control site (typically 180° from target) at the same time and expressed those as a ratio.

*4) Untested assumptions:*

*The narrative makes assumptions that are not directly tested. Articulating and justifying each assumption (with citations or references to salient experiments, or with some appropriate rationale) would be very helpful:*

*(i) It is assumed that more frequent light pulses result in greater recruitment of ePDZ fusions.*

We include additional results to substantiate this point. We provide a dose response of mCherry-ePDZ recruitment to different intensity regimes in Figure 1—figure supplement 2. The decrease observed at the highest intensity levels is not due to photobleaching as the GFP signal did not exhibit photobleaching, but rather may be due to light-induced rupture of the LOV2-flavin mononucleotide adduct (Kennis et al., 2004}.

*(ii) It is assumed that mutant ePDZ fusions are expressed at similar levels to WT fusions.*

We explicitly state this assumption in the Materials and methods section.

“Cdc24 and Bem1 point mutants were assumed to accumulate at levels comparable to the wild-type proteins; most mutants were previously characterized

(Ito et al., 2001; Irazoqui et al., 2003; Butty et al., 2002), the Dbs:Cdc42 co-crystal structure (Rossman et al., 2002) was used to choose point mutations to inactivate Cdc24 GEF activity.”

*(iii) It is assumed that illumination has no effect except to recruit ePDZ fusions.*

This has been tested as no polarization or other response is seen in control cells in which mCherry-ePDZ control is recruited. This is also substantiated by the observations that recruitment of GEF-dead Cdc24 or Bem1 K482A do not induce polarization or other response.

*5) Description of feedbacks:*

*The text should also be more precise in the description of feedbacks. The authors show data that is consistent with positive feedback, but do not directly demonstrate positive feedback: recruitment of Bem1 by Bem1 is consistent with feedback, but equally consistent with oligomerization, and as highlighted above, Cdc24 recruitment by Cdc24 could also be due to oligomerization. The text should be clear on these alternative possibilities.*

We have performed additional experiments to address this point. To rule out oligomerization, we examined whether GEF-dead Cdc24 could recruit Cdc24-tdTomato and whether Bem1 K482A could induce Bem1-tdTomato. In both cases, no recruitment was observed. This substantiates that we are detecting bonafide positive feedback.

*Similarly, regarding the proposed pre-start feedback, there is no direct data showing Bem1 independence. While optogenetic recruitment of Cdc24-ePDZ does not lead to detectable recruitment of endogenous Bem1, absence of evidence is not evidence of absence. Without testing that Bem1 is indeed not required, the wording of the text should be changed to make this clear.*

In the text we state, “These data indicate that optogenetically-initiated sites of Cdc42 activation are stable and are maintained, at least in part, by the accumulation of Cdc24, without accumulation of detectable Bem1.” And throughout this section we state that this occurs in the “apparent absence of Bem1”. Thus, we highlight the extent to which Bem1 is recruited to detectable levels.

We added the following to the Discussion to acknowledge that the Bem1 independence of early activation of Cdc42 has not been directly tested.

“Synthetic lethality precluded generation of a conditional Bem1 allele compatible with our experimental approach which would allow a direct test of the Bem1 independence of Cdc24-mediated activation of Cdc42 during early G1.”